# Design and Implementation of Fuzzy Compensation Scheme for Temperature and Solar Irradiance Wireless Sensor Network (WSN) on Solar Photovoltaic (PV) System

**DOI:** 10.3390/s20236744

**Published:** 2020-11-25

**Authors:** Abdul Rahim Pazikadin, Damhuji Rifai, Kharudin Ali, Nor Hana Mamat, Noraznafulsima Khamsah

**Affiliations:** 1Faculty of Engineering Technology, TATIUC, Kemaman 24000, Terengganu, Malaysia; damhuji@uctati.edu.my (D.R.); kharudin@uctati.edu.my (K.A.); norhana@uctati.edu.my (N.H.M.); sima@uctati.edu.my (N.K.); 2ZSR Vortices Sdn. Bhd. Prima Dagangan, Bandar Baru Kijal 24100, Terengganu, Malaysia

**Keywords:** photovoltaic solar systems, monitoring, solar power generation, wireless sensor network

## Abstract

Photovoltaic (PV) systems need measurements of incident solar irradiance and PV surface temperature for performance analysis and monitoring purposes. Ground-based network sensor measurement is preferred in many near real-time operations such as forecasting and photovoltaic (PV) performance evaluation on the ground. Hence, this study proposed a Fuzzy compensation scheme for temperature and solar irradiance wireless sensor network (WSN) measurement on stand-alone solar photovoltaic (PV) system to improve the sensor measurement. The WSN installation through an Internet of Things (IoT) platform for solar irradiance and PV surface temperature measurement was fabricated. The simulation for the solar irradiance Fuzzy Logic compensation (SIFLC) scheme and Temperature Fuzzy Logic compensation (TFLC) scheme was conducted using Matlab/Simulink. The simulation result identified that the scheme was used to compensate for the error temperature and solar irradiance sensor measurements over a variation temperature and solar irradiance range from 20 to 60 °C and from zero up to 2000 W/m^2^. The experimental results show that the Fuzzy Logic compensation scheme can reduce the sensor measurement error up to 17% and 20% for solar irradiance and PV temperature measurement.

## 1. Introduction

Malaysia is known as a country with unique geographical conditions and consists of the Peninsular Malaysia, Sabah, and Sarawak. Malaysia’s northern part of Borneo faces a more challenging situation in the reformation of religious schools in rural areas. Schools in this part are situated in areas that are accessible only by a limited-access road and are very difficult to reach for electricity and utilities. This situation significantly affects the school revitalizing process. Many private electricity utilities providers are reluctant to electrify remote rural areas because it is too costly to string electrical lines with low load factors into these inaccessible sections. There are over 1000 religious schools in Malaysia where 30% of these schools are in rural areas and far from electricity facilities [1]. The school of Tahfiz Ribat Al-Asyraf Pondok Mak Lagam is located approximately 30 km from the nearest town and is outside the grid of Tenaga Nasional Berhad (TNB) electricity network. Electricity sources for this school are provided using generators. This leads to a high consumption of oil and the noise generated by electric generators hinders students’ learning process. Therefore, renewable energy from solar PV is viewed as the best alternative to replace electric generators.

The Malaysian Ministry of Energy and Technology has stipulated a target of 20% of the country’s electricity to be produced from renewable energy by 2030 [2]. The increased implementation of a solar photovoltaic (PV) system as an alternative to conventional generators as an electric source has revolutionized the conventional electricity utility planning of power management. An efficient method of power generation forecasting and energy accumulator of solar PV power is crucial for producing an economical balance of the operations of the power systems [3,4].

Various factors affect the accuracy of PV solar power generation forecasting such as the state of the climate parameters and horizon time of the forecasting. Poolla et al. [5] identified that the error between forecasting and measurement of power generation is the combined effect of error modelling and sensor error in measuring the weather conditions. The accuracy of weather measurement is known to be intermittent, and a sudden and large unexpected variation can negate a forecast. The accuracy sensor measurement of solar irradiance intensity and the temperature are factors that influence the forecast accuracy of solar PV power generation [6,7]. 

The literature has reported that the problem with many sensor measurements is that the accuracy of these measurements is mainly affected by the variation of the surrounding weather. For temperature measurement on the surface solar panel, the problem occurred with the position of the sensor under the solar panel. Kuitch et al. [8] proved that the location of the temperature sensor at the back of the solar panel cell introduced an error temperature of between 2 and 15 °C. Statistical analysis in this study suggests that correction compensation is needed to estimate PV cell surface temperature. However, no suggestions and conclusions were proposed, as the research is in its preliminary stages. The foregoing analysis reported that the distance of the sensor to the surface solar panel and the variation surrounding changes were the main factors that contribute to dynamic sensor PV cell temperature error measurement [9,10]. 

Other factors that affect the temperature sensor measurement are the wind speed and surrounding humidity. For solar irradiance and wind sensor measurement, the existence of sensor error was mainly due to variation changes in the surrounding thermal expansion, and long cable sensors will cause a voltage drop [11]. The voltage drop is attributed to the resistance of wires and the effect on the active sensor is more critical, as there is more current streaming of active sensors in the ground wires [12]. 

Recently, Abdul et al. [7] gave a comprehensive review on instrumentation for irradiance measurement. They suggest that the precision of solar power generation forecasting primarily depends on the accuracy of solar irradiance measurement and the proper calibration of a solar irradiance sensor will improve the measurement precision. The finding is consistent with findings of past studies by Vignola et al. [13], which demonstrated that the intensity of solar irradiance has the highest influence in solar power generation. This is supported by the Olano et al. [14] study, which reveals that the measurement of solar irradiance using a pyranometer sensor can improve through proper calibration under three types of cloud condition. They also suggested that the statistical error compensation for pyranometer sensor measurement establishes high accuracy solar irradiance measurement.

A compensation technique is needed to improve sensor accuracy measurement by eliminating the sensor error measurement. This technique enhances the persistence of power PV generation forecasting [15,16]. To the author’s best knowledge, previous studies on sensor error compensation have only focused on a single type of sensor. Therefore, the motivation for this study is to design and implement smart sensor error compensation for multi-sensors using smart Fuzzy Logic algorithms for the PV power generation plant system. The enhancement in the accuracy of solar irradiance intensity and PV cell surface temperature measurement will improve the solar power generation forecasting and PV assessment monitoring that would be highly beneficial to the system operators for an efficient planning and operation process.

## 2. Stand-Alone Photovoltaic System

Many religious schools in Malaysia located in rural areas do not have access to the national grid. Hence, a PV power generation plant system is a practical and cost-effective alternative electrical energy source. A PV system is easier to operate than a diesel generator as compared to an alternative electrical energy source. This system is capable of reducing running costs, requires low maintenance, and exerts minimal environmental impact [17]. The last few years have witnessed the significant maturity of PV technology, which enabled the standardization of modular components. Market patterns indicate a substantial decline in costs and an increase in foreign PV production at 33% annual compound growth [18]. There is a significant demand for energy-efficient lighting and appliances in the international markets. This may help maintain the competitiveness of PV systems due to the lower load and reduction in the size of the PV system. The PV modules are mounted in arrays configuration and connected to the frame. The PV module consists of thin semi-conductor material cells, which transform radiation from the sun into electricity from DC. The panel is coated with transparent material, which is sealed for proofing water. This system uses a monocrystalline solar panel with 72 cells and a 330 W capacity. Solar panels often prove to be more effective at lower temperatures which require them to be installed in a way that enables free air passage around them. Bright cool days are the perfect conditions for PV systems to produce maximum energy [9].

A stand-alone PV system is an electrical system consisting of PV array modules that are usually of 12 volts with output between 50 and 100 watts each. The PV modules used in this PV system are monocrystalline 72 cells that produce a maximum power output of 330 W at irradiance 1000 W/m^2^ and a cell temperature of 25 °C. The electrical parameters of the PV panel for I_PV_ and V_PV_ at short circuit, open circuit, the number of the PV cells, and the maximum power point are shown in Table 1.

Then, these PV modules are combined into a single array to deliver the required output power. A stand-alone PV system is a solar panel that generates electricity to charge the battery bank during the day for the night use when the energy from the sun is not available. Figure 1 shows the PV solar stand-alone systems block diagram for this project. PV systems transform sunlight directly into electricity without fuel or maintenance and generate renewable energy without pollution. The amount of power that a PV system can capture depends on the strength of the solar irradiance, the temperature of the surface panel, and the efficiency of the mounted PV system installed.

The effectiveness of an off-grid PV system depends on the efficiency of the batteries. The batteries must be charged correctly and held at a high charging level for batteries to operate well and have a long service life. A charging controller monitors the power from a PV module to prevent batteries from overloading and produce high voltage levels. 

A charge controller with maximum power point tracking (MPPT) will control the photovoltaic output voltage to obtain the maximum power. PV array maximum power point tracking is essential to deliver the maximum available power from the PV cells to the load. The stand-alone solar system is operated at the maximum power point condition with a total maximum current I_Max_ = 70.9 A. The solar system with MPPT topology should provide an output power of 2.5 KW to meet the daily load requirement of the school. The desired output voltage and current with the MPPT should be 37.5 V and 8.78 A respectively to fulfill the power demand throughout the day. 

The maximum power point tracking (MPPT) used for this project is SMART1 60 A manufactured by BLUE SUN SOLAR. This model is an enhanced/hybrid smart solar charge controller that is incorporated with advanced maximum power point tracking (MPPT) technology. An essential aspect of the off-grid solar system is the solar charge controller. The controller is capable of tracking 99% of the peak power with efficiency conversion while using the advanced maximum power point tracking (MPPT) technology. Inside the controller, the MPPT microprocessor generates 30% more charging current, which is slightly more powerful than the conventional technologies. Other benefits include faster deployment and volume expansion support. These can store energy for different types of batteries. The smart system controller can also be upgraded online to provide a lifetime update service to the system. This system can work in high temperatures as high as 105 °C.

Another important part of this PV system is the energy storage system. In stand-alone photovoltaic systems, the lead–acid batteries are almost exclusively used. Although different battery types have advantages such as high storage capacity or lower self-discharge, the definitive advantage of the lead battery is its low price. Batteries reflect the largest expense of components over the lifespan of a stand-alone solar PV device. 

A battery used in a stand-alone solar photovoltaic system is charged during daylight and can withstand deep discharge during no-sun hours. These batteries are called deep cycle batteries. Differing specifications impact the two different forms of building. Batteries should be of the same type and manufacturer within a battery bank, and about the same age. Old or poorly performing batteries impact the reliability and reduce the performance.

Inverters in this PV system are expected to be able to support critical loads as uninterruptible power supplies. The inverter’s principal role is to transform DC to AC electricity. The inverter has to meet two main criteria, which are maximum surge power and continuous power. Figure 2 illustrates the complete installation of the stand-alone solar system.

### Mathematical Modeling Relationship of PV Array to Temperature–Solar Irradiance

An initial review of the relevant research revealed detailed prior work on identifying the important influence of weather on solar power generation. Numerous studies have shown that the solar irradiance intensity and surface temperature of the PV modules are the most influential weather parameters on solar power generation forecasting. This section will derive the relationship between the temperature solar PV module and solar irradiance intensity in the solar PV module. PV arrays are composed of a series or parallel combinations of PV solar cells, which are usually represented by a simplified circuit equivalent model, as shown in Figure 3 [19,20].

The PV cell output voltage is a photocurrent feature that is mainly determined during the process by the load current, depending on the degree of solar irradiation. The correction coefficients used to change the voltage *V_C_* and current *I_ph_* to specific PV module temperature and irradiance values are as follows [13]: (1)$$V_{c}=\frac{AKT_{c}ln}{e}ln\left(\frac{I_{ph}+I_{o}-I_{c}}{I_{o}}\right)-R_{s}I_{c}.$$

Temperature correction factors for voltage and current respectively are: (2)$$C_{TV}=1+\beta_{T}\left(T_{c}-T_{x}\right)$$
(3)$$C_{TI}=1+\frac{\gamma_{T}}{S_{c}}\beta_{T}\left(T_{c}-T_{x}\right).$$

Irradiance correction factors for voltage and current respectively are: (4)$$C_{sv}=1+\beta_{T}\alpha_{T}\left(S_{x}-S_{c}\right)$$
(5)$$C_{si}=1+\frac{1}{S_{c}}\left(S_{x}-S_{c}\right).$$

Applying the correction factors to *I_ph_* and *V_c_*, the temperature–irradiance relationship was obtained.
(6)$$V_{cx}=V_{C}C_{TV}C_{SV}$$
(7)$$I_{PHX}=C_{TI}C_{SI}$$
where the symbols are defined as follows:*e*: electron charge (1.602 × 10^−19^ C). *K*: Boltzmann constant (1.38 × 10^−23^ J/K). *I_c_*: cell output current, A. *I_ph_*: photocurrent, function of irradiation level, and junction temperature (5 A). *I_o_*: reverse saturation current of diode (0.0002 A). *R_s_*: series resistance of cell (0.001 Ω). *T_c_*: reference cell operating temperature (20 °C). *V_c_*: cell output voltage, V.*A*: curve fitting factor. *T_x_*: PV module temperature (variable), K.*S_c_*: STC irradiance, W/m^2^*S_x_*: variable irradiance, W/m^2^.*C_TV_*: temperature voltage correction factor. *C_TI_*: temperature current correction factor. *C_SV_*: irradiance voltage correction factor *C_SI_*: irradiance current correction factor*α_T_*: slope of change in the cell operating temperature due to change in the solar irradiation level (Km^2^/W)*β_T_*: correction coefficient (1/K)*γ_T_*: correction coefficient (w/Km^2^)

The characteristic graph of the PV cell is visualized in Figure 4, which depicts that the V_OC_ is an open circuit voltage of the cell, I_SC_ is the short circuit current, and P_max_ is the maximum DC power for certain temperatures and sun irradiation.

## 3. Solar Irradiance and PV Surface Temperature Sensor System

PV powers generating forecasting is usually neglected by solar farm power generation companies. However, the current research trend proves that solar power generation forecasting is crucial and improves the electrical power management. The temperature and solar irradiation are the main factors that affect and contribute to the number of power generators, as shown in Equation (1). Countries situated on the Equator such as Malaysia receive a daily average of irradiation is 7.7 kW/m^2^ with rainy monsoon from September to December. The rain monsoon reduces the intensity of solar irradiation and the PV modules’ temperature, consequently affecting the amount of electricity generation [22].

This motivates the development of an IoT-based wireless network PV modules system to build a monitoring network for PV modules suitable for large-scale solar farm applications [23]. Fault detection is also another justification for the monitoring of individual PV modules. This definition is based on cost savings, modularity, and practicality of installations, since hundreds of modules may be required for PV plants [24,25]. 

The main objective of the sensor network is to measure and record the solar irradiance and PV surface panel temperature is the baseline of IoT. The implementation of IoT is to fulfill the demand for the increasing amount of data to conduct accurate solar forecasting. The architecture of the solar irradiance and temperature wireless sensor network (WSN) system for characterizing the simultaneous effects of temperature and solar irradiance on power solar generation is presented in Figure 5. The sensor system aims to provide irradiation and temperature measurement for solar power generation characterizations and model validation studies. The main objective of the ground base system is to obtain a fundamental measurement for characterizing solar resource magnitude in the Kemaman district for its potential application as an alternative energy source in rural areas.

A solar irradiance sensor Si-mV-85 and digital temperature sensor DS 18820 were used to measure the intensity of the irradiance and temperature of the solar panel. These are measured on both the upper and below the solar panel surface. Numerous practical wireless sensor networks include sensor nodes, a sink node, an internet link, and a node handling function. There is no fixed location for the sensor nodes, so most are randomly deployed to track a sensor area. The sensor nodes used a multi-hop method to communicate with each other through a wireless on-board Wi-Fi. The data obtained from the sensor field are sent to a base station (sink) after primary processing, which is responsible for transmitting data to another network. It features and renders a sink in a conventional network, which is equivalent to a gateway. Finally, the measuring data of solar irradiance and temperature enter the task manager node and are accessible to the users. 

For commercial applications on a large energy solar farm that contains thousands of solar panels, the WSN comprises thousands of sensing devices. These sensing devices are deployed in a sensing field in a specific location that optimizes the number of sensing devices and the accuracy of the solar panel temperature and irradiance measurement that cover the entire solar farm. The sensors transmit the measuring data through specialized sensors and send them to the base stations (BS). A sensor device mainly consists of four parts, which are the sensor, transceiver and transmitter, a primary processing unit, and a low power source. Some WSNs also have elements of a stand-alone power generator and location-identifying unit. The structure of the IoT base wireless sensor network for PV solar panel characterization is shown in Figure 6.

The sensing unit consists of three sub-units, which are the temperature sensors, the solar irradiance sensors, the wind velocity sensor, and the analog to digital converters (ADC) integrated into the ESP module. For this system, the Arduino acts as a sensor node that functions as a field controller in a large wireless network sensor. The schematic circuit of the system is designed and simulated using Fritzing software. The power module for the sensor node power supply consists of a 5 V_DC_ and 12 V_DC_. The main supply of 220 V_AC_ is supplied by the solar farm system. Figure 7 shows the connection of the electronic component of the node sensor, which consists of a power converter Arduino Mega, digital temperature sensor, anemometer sensor, and irradiance sensor. Sensors sense the parameters to measure and the ADC converts the sensed data to digital signal form to allow the sensor node to perform preliminary processing. The transmitter and transceiver port enable the connection of the sensor nodes to the Internet networks. The most important part of these WSN nodes is its power unit, which is a 12 V_DC_ and 5 V_DC_ provided by devices with sensor and sensor nodes. 

All of the electronics components and sensor modules for the wireless network sensor are installed in a PVC junction to protect the system from the fluctuating environment temperature that affects the electronics board and prevents a short circuit from live wiring that can damage the system. A power supply module of 12 V_DC_ was used to convert AC voltage to DC voltage. The DC output from the 12 V_DC_ supply module is stepped down to 5 V_DC_ using the DC converter 12 V–5 V and connected directly to the Arduino Mega, ESP8266 Wi-Fi module, Irradiance Sensor, Anemometer Sensor, and Digital Temperature Sensor DS 18820. Figure 8 demonstrates the full sensor node electronic system installation for the WSN PV solar panel monitoring. The node sensor box is mounted on the wall of a room at Surau Pusat Pengajian Islam Ribat Al-Asyraf, Kemaman.

The WSN PV module system works in parallel to simultaneously send data from each sensor every 10 s for 24 h daily. These sampling intervals data comply with the IEC 61724:1998, which requires the interval data sampling to be less than 1 min.

The sensors used in this research are a sensor Si-mV-85 for solar irradiance measurement and a DS 18820 digital temperature sensor for measuring both below and above the solar module surface. The temperature surface measurement of the solar panel is used as a reference for measuring temperature below the solar panel. In a real application, many researchers install the temperature sensor below the solar panel. This affects the accuracy of measurement, since the taken measurement reflects the real surface on the upper side of the PV panel. Initial study shows the error highest up to 10 °C which reflects a 20% error. The installation of the temperature sensor on the PV surface panel and irradiance sensor is shown in Figure 9. This location is able to ensure an accurate reading of the solar irradiance that closely reflects the solar irradiance that coincides to the panel of the PV module.

Silicon irradiance sensors (Si sensors) provide a cost-effective, robust, and reliable solution for measuring solar irradiance, notably photovoltaic (PV) monitoring systems. Ideally suited as a guide for monitoring PV systems, they are based on the construction of sensors corresponding to a PV module. Specifically, the spectral responses are comparable to PV modules, and also, the similar inclination error (incident angle modifier) allows an accurate analysis of the PV energy yields using Si sensor data.

The silicon irradiance sensor (Si-mV-85) is used to measure the natural solar irradiance up to 1500 W/m^2^, the operating cell temperature is −35 to 80 °C, and it has an accuracy around ±5 W/m^2^, and also the output voltage of this sensor is approximately 85 mV for 1500 W/m^2^. It has a maximum output voltage signal of 85 mV. Thus, it directly connects to the data logger without any signal conditioner. 

Since ESP8266 has only one analog input, the irradiance sensor is connected to the Arduino’s analog input. Using the Arduino Analog-to-Digital Converter (ADC) provides 10-bit analog voltage readings. It means any reading of voltage between 0 and 3.3 V can be associated with a value between 0 and 210, or 1024. A higher resolution external ADC provides more precise readings. However, since the Si-mV-85 irradiance sensor has an accuracy of ±0.05 under optimum conditions, the precision level from the 10-bit ADC is adequate. Several consecutive samples may be taken and averaged to improve the accuracy of the ADC readings and reduce the impact of electrical noise by surrounding components. Figure 10 shown the installation of the node sensor for WSN.

## 4. Calibration of Sensor Measurement

This section described the laboratory setup for sensor calibration measurement of the WSN temperature and irradiance sensor system. This measurement was carried out to collect data from the sensor for accuracy analysis of the WSN system to monitor the PV panel. The objective of this analysis is to indicate the accuracy and error reading of the WSN sensor module system. Smart error compensation based on the Fuzzy Logic algorithm is proposed to improve the measurement of the WSN system. Preliminary calibration measurements of the sensor module for data collection and preliminary analysis were conducted in Sensor Lab at UC TATI.

The performance and accuracy of the temperature and solar irradiance sensor module and WSN data loggers were first tested alongside commercial sensors in laboratory conditions followed by testing on the PV solar panel field. An SM206 High Precision Solar Power Meter Light Meter was used to calibrate the solar irradiance sensor. These commercial devices were chosen for comparison because they are widely used in both the industry and academic research. Measurement and calibration were conducted in two different conditions, which are the laboratory and location site of the PV panel system to ensure the accuracy of sensor measurement of the WSN system. Data collection recorded an interval time of 30 s with control parameters. Figure 11 shows the configuration of the calibration process for irradiance and temperature module in the laboratory.

The heat bar was used as a platform for temperature sensor calibration. Standard industrial temperature digital thermometer measurement was used as a reference to analyze and compare the accuracy of DS18B20 temperature sensor measurement. The heat bar is a means by which a range of temperatures can be established and is illustrated in Figure 12.

The heat bar consists of three heating elements, two of which are used in series or parallel for line voltages of 220 V or 110 V. The third “aux heater” element on the control box panel is a low-power one, which is terminated separately in two taper sockets. The bar itself transmits heat from the heater through conduction to the heat sink. It is marked off with notches at intervals of 1 cm for the easy placement of different transducers along with it. The heat sink transmits heat to the surrounding air through convection from the bar. Therefore, the cool bar end is just slightly above room temperature.

The DS18B20 temperature sensor and industrial standard digital thermometer are immerse in a tank that contains water, which functions as a heat transfer medium. The tank is mounted on a heat bar that is centrally located over the heat bar notch. After several minutes, reading of the indicated temperature is recorded as an initial temperature, and then the heat bar power supply is switched on to increase the temperature of water on the tank. The temperature reading is recorded every 2 min for the first 20 min and every 5 min until no further increase is observed.

The DS18B20 temperature sensor output voltage was sent to the PC via a data acquisition (DAQ) system for data collection, post-processing, and measurement display. To acquire the signal, serial interfacing for DAQ used the Arduino Mega, and the data collection and logging were completed using the Thinger IO platform as previously mentioned. The Arduino data interfacing collected data at a speed of 1000 Hz, which sampled 500 data every 1 ms.

The calibration of the irradiance sensor Si-mV-85 was conducted by comparing the measurements with an SM206 High-Precision Solar Power Meter Light Meter at the laboratory under a 500 W tungsten-halogen lamp and at the field location of the solar panel such as shown in Figure 9. The irradiance intensity variation is achieved by varying the distance of the irradiance sensor to the radiation source. Experimental verification was performed by varying the distance between the irradiance sensor Si-mV-85 and the radiation source from 100 to 300 cm. The laboratory calibration is verified by comparing the measurement of the irradiance sensor Si-mV-85 with the SM206 meter located on the PV panel PV. This method also verified that the default calibration distance of 1500 mm measurements is consistent with the measurements obtained at the actual position of the PV panels. 

Low irradiance intensity significantly increases the noise of measurements due to the massive distance of the sensor from the source. Such tests also demonstrate the robustness of the instrument under strict environmental conditions in countries on the Equator line, as the instrument was deployed outdoors. The calibrations were carried out on 3 January 2020 and 9 January 2020, which can also establish regular laboratory calibration references. 

## 5. Sensor Accuracy Analysis

The literature has identified that the problem with many sensors is the accuracy of measurements, which is greatly affected by the variation of the surrounding temperature. For temperature measurement on a surface solar panel, the problem occurred with the sensor placement under the PV panel. Poolla et al. [5] proved that the location of the temperature sensor at the back of the solar panel cell introduced an error temperature of 2 to 20 °C. Statistical analyses suggest that the correction factor needs to add to the temperature measurement algorithm. Foregoing analysis reported that the distance between the sensor and surface PV panel and the variation surrounding changes were important factors that contribute to the dynamic sensor error measurement. 

For solar irradiance sensor measurement, the existence of a sensor error was mainly attributed to the variation changes in the surrounding thermal expansion and voltage drop on long cable sensor wires. The voltage drop follows Ohm’s law and induces a noticeable rise in stress between the signal lead and the signal reference lead. This voltage drop is due to the resistance of wires and the effect on the active sensor is more important, as there is more current flowing through the active sensor’s ground cables. A compensation technique is needed in order to improve the sensor accuracy measurement and improve the forecasting of power PV generation.

According to Van Hees et al. [26], the thermal sensor has limitations of measurement errors associated with mean temperature drift. The relation between the sensor measurements and environmental temperature shows that the surrounding temperature is an important factor influencing the accuracy of temperature sensor measurement. Therefore, for a real working environment in which temperature variation is high, the error is drastically increased. 

The sensor reading on the upper side of the PV panel was selected as the standard or reference temperature in this study. Based on the selected standard temperature, the absolute error (e) of the measurement is as shown in Equation (8).
(8)$$\left|e=T_{b}-T_{u}\right|$$
where *T_b_* is the temperature measurement below the PV panel, and *T_u_* is the temperature measurement above the PV panel. Hence, the percentage error in the measurement is as shown in Equation (9).
(9)$$\%e=\left|\frac{T_{b}-T_{u}}{T_{u}}\right|\times 100\%$$

The absolute error and percentage of error for each irradiance and temperature variable were calculated based on a commercial meter as a reference value. The calculated absolute error and percentage error are shown in Table 2 and Table 3. The measurement temperatures and solar irradiance were taken from 7:30 a.m. to 7:30 p.m. to produce an error from 0.2422% to 33.3421%, as shown in Table 2 and Table 3. The data analysis shows that the error increases with an increase in temperature and intensity of the solar irradiance.

## 6. Fuzzy Irradiance and Temperature Compensation Scheme (FITCS)

Based on the accuracy analysis in Table 2 and Table 3, and in order to meet the measurement precision, the measured data must be compensated, since the error of irradiance and temperature is too high. A Fuzzy Irradiance and Temperature Compensation Scheme (FITCS) was proposed to compensate the measurement error caused by the variation of irradiance and temperature using a Mandani-type fuzzy inference system, as shown in Figure 13.

The Mamdani fuzzy inference was implemented to develop a control system by synthesizing a collection of language control rules obtained from experienced human operators. The output of each rule is a Fuzzy set in a Mamdani scheme. Even though Mamdani systems are more intuitive and easier to understand, they are suitable for expert system applications where the rules are developed from human expert experience [27].

The proposed method allows the physical input parameters to be interpreted directly after calculating its value and compensated for the irradiance and temperature changes. The actual sensor and the Fuzzy compensation scheme are related to cascade. In this scheme, the sensor output is fed as the system inputs, and the error is corrected based on the proposed rules.

In this scheme, the FITCS has two inputs, which are the variations of irradiance and temperature. The output feedback of the FITCS needs to be done for the sensor output. Matlab’s Fuzzy Logic Toolbox was used to assist in the development of FITCS. The toolbox contains functions, user interfaces, and data structures that allow an inference system to quickly design, test, simulate, and alter a fuzzy inference system.

Figure 14 illustrates that each input is associated with a Fuzzy set and each Fuzzy set accepts membership functions (MF). The MF reacts to the degree of each Fuzzy set of the membership on a scale of 0 to 1. The Fuzzification is carried out to accompany the Fuzzy set with the MFs. Fuzzy rules are stated in IF–THEN Lingual Convictions, which describe the relationship between input and output.

The Fuzzy inference system was developed for an error compensation technique system using Mamdani-type Fuzzy models, which are based on irradiance and temperature measurements. Fuzzy Logic IF used in general cases such as to handle vague and imprecise information is based on linguistic variables. The Fuzzy sets include the information that is combined with rules to identify the action to be taken. 

This Fuzzy-based decision-making system contained a system input, system output, membership functions (MF), and IF–THEN Fuzzy rules. The inputs were features of irradiance and temperature measurements. The output of the system was the actual measurement of irradiance and temperature values. Each input was related to one Fuzzy set, and each Fuzzy set has its own corresponding membership function (MF). The MF responded to the degree of each Fuzzy set as a member in the membership on a scale of 0 to 1. 

Fuzzy Logic analysis was applied as an administrator to provide the irradiance and temperature data. In particular, this Fuzzy-based deciding scheme contained a system input, system output, membership function (MF), and IF–THEN Fuzzy rules. The inputs referred to the amplitude of the irradiance and temperature sensor. As shown in Figure 14, each input is associated with one Fuzzy set with an agreeing MF. The MF reacted to the degree of each Fuzzy set as a member of the membership on a scale of 0 to 1. Furthermore, it should be noted that the Fuzzification was executed appropriately as the companion of the Fuzzy set with MFs. Fuzzy rules were declared in IF–THEN lingual condemnations whereby the relative between input and output could be described as follows: IF the temperature and times (input) was high, the actual temperature is high (output). Eventually, a defuzzification action was required to convey the lingual variables into crisp mathematical values for more than one Fuzzy rule, which was consistently applied.

In this study, irradiance and temperature data were measured and tested to investigate the error and actual value of the sensor output. For each MF, the number of MFs and the Fuzzy rules must be developed independently according to different feature groups. In this work, the rule algorithm was utilized as the system learning process to obtain the Fuzzy Logic system. Subsequently, the trained Fuzzy Logic engine was applied to compensate the irradiance and temperature data based on the extracted features.

## 7. Simulation Model of Fuzzy Sensor Compensation Scheme

For the FITCS simulation, a sensor compensation scheme model was developed using the Fuzzy Logic Toolbox to validate the proposed scheme. The model is constructed using the Fuzzy Logic Toolbox, as shown in Figure 15. The input to the model scheme is the variation of temperature and irradiance. The output of the Fuzzy Logic controller is the correction needed to be done according to the measurement of the sensor. The Fuzzy Logic controller was also constructed based on the proposed scheme. According to the fifteen rules developed in the proposed scheme, Fuzzy Logic controller systems are used in this model. The output after compensation is displayed using an oscilloscope and analyzed to validate the effectiveness of the proposed scheme.

The internal block function in FIS is shown in Figure 15, and the input features were temperature signals, irradiance signals, and the known output was the compensate temperature and irradiance data. Using a hybrid learning rule with linear output, a Mamdani Fuzzy Logic system with multiple inputs single output (MISO) was generated. 

### 7.1. The Fuzzy Inference Steps

Fuzzy inference is the approach used to formulate the mapping from a given input to an output using Fuzzy Logic. The Mamdani-type Fuzzy Inference System (FIS) is based on five steps which are normalization, Fuzzification, determination of Fuzzy rules (Fuzzy inference engine), defuzzification, and demoralization [28]. The normalization was achieved by utilizing the information from the temperature and irradiance sensor, which was obtained from the calibration of the temperature and irradiance sensors. Temperature measurement that is usually practiced by electricity and utility companies is done by locating the sensor under the PV panel, which does not reflect the real surface of the PV panel. The analysis in Table 2 shows that the error is up 20% when the surface panel temperature more than 50 °C. The characteristics of different temperature measurements show a nonlinear variation of error. During the day, the temperature measurement of the PV surface panel is higher than the PV back panel. This circumstance was quite obvious, especially for the range of temperatures 40–60 °C during the day and range temperature of below 24 °C during the night. 

The next step was to convert the crisp peak amplitude value of the irradiance and temperature sensors into a linguistic variable (Fuzzy value), and this process is known as Fuzzification. For this FIS, three linguistic variable inputs have been developed according to the peak amplitude values of the temperature and irradiance sensor. These values were read and must be normalized to the range of 20 to 45 according to the minimum and maximum temperature and the intensity of irradiation, which was carried out appropriately to meet the companion membership function with the Fuzzy sets. Each linguistic value was assigned a Gaussian membership function for each of the two input variables. The membership functions for the irradiance and temperature sensors signals are shown in Figure 16, and these functions were the same for both input variables.

Once the input data were normalized and Fuzzified, the Fuzzy inference was used to build up the Fuzzy rules. These rules were built depending on the requirement of the system, and after its execution, their membership function was determined in relation to the output set. Fuzzy rules were stated in IF–THEN lingual condemnations, which described the relationship between the input and output. As mentioned above, the experimental trial started with the MISO inference model, where input variables irradiance and temperature signals were represented by Fuzzy sets. The Fuzzy rules were developed based on the actual situation and experience, instead of the availability of the system model. The structure of the general rule can be written as:IF e(k) IS W AND Ae(k) IS Q THEN Au(k) IS C
where Au(k) was the change in Fuzzy input (which was the output of the Fuzzy Logic) and W, Q and C were the Fuzzy sets defined over the universe of discourse e, Ae and Au respectively, with the linguistic variables used for the output signal Au(k). The membership function for the output variable Au is depicted in Figure 17.

Finally, the defuzzification step was applied to transport the lingual variables into crisp mathematical values. Through this process, all rules were searched to identify the rule value vector that provided a Fuzzy value for a particular output variable. When each rule described a particular output, its rule value was used to change the corresponding membership function. The new crisp value presented the compensated signal output of the temperature sensor, which was an accurate reading of the surface PV panel temperature. A Fuzzy conclusion of “temperature is *A* and irradiance *B*” can be determined by Equations (10) and (11) in the discrete domain.
(10)$$y=\frac{\sum_{i}\mu A(y_{i})\times y_{i}}{\sum_{i}\mu A(y_{i})}$$
(11)$$y=\frac{\sum_{i}\mu B(y_{i})\times y_{i}}{\sum_{i}\mu B(y_{i})}$$
where *µA(yi)* and *µB(yi)* was the membership function. Having defuzzified the data, their values were denormalized to obtain the real value, which was used as the new compensate temperature and irradiance data.

### 7.2. Fuzzy Simulink System

The Fuzzy Logic Toolbox is designed to work seamlessly with Simulink, the MathWorks simulation program. When the Fuzzy system has been developed, it is ready to be directly implemented into a simulation. Thus, once the model is created in a Fuzzy Logic Toolbox, it should be exported to a workspace so that Simulink automatically recognizes it and builds the model.

The MATLAB/Simulink Toolbox was utilized to process the temperature and irradiance input signals with the Fuzzy inference system and the output signal. Figure 18 and Figure 19 show the simulation block that was used to integrate the temperature and irradiance data with Matlab software for further signal analysis in which two signals were measured by the sensors. Then, they were treated as inputs for the error compensation technique, which was done using Mamdani FIS. The simulation model consisted of three parts, which were utilized for the error compensation technique. The first part was the two outputs of temperature and irradiance sensors, which were set as inputs. The second part was the Fuzzy Inference System (FIS) model, being used to compensate for the errors. Then, the Fuzzy block setting was exported to the Matlab workspace to support the Simulink. The feedback and the error compensation would be processed according to the output signal from the Fuzzy Logic. The output signal from the Fuzzy block will be brought through the error compensation equation in the signal block, and it would be fed back into the error block, and lastly forwarded into the output display.

## 8. Experimental Result and Discussion

In order to investigate and monitor the performance of PV solar power generation, the proposed WSN based on the IoT system is constructed to monitor the solar irradiance intensity and PV panel surface temperature that affect the power solar PV generation. The main goal of this research is to compensate for the sensor measurement error that was introduced by technical and environmental factors. The error of temperature PV panel monitoring is caused by the placement of the sensor under the solar panel that does not reflect the actual temperature surface of the PV panel. For solar irradiation measurement, the existence of sensor error was mainly due to the variation changes of the surrounding thermal expansion and voltage drop on a long cable sensor wire. 

Recording sufficient temperature and solar irradiance is important for post-processing compensation. The solar irradiance is directly recorded by the WSN through the IoT platform and storage on the IoT Thinger drive. The data are downloaded and go through preprocessing selection data. The invalid data that are affected by noise are removed in order to obtain the real condition of the PV module. The system is run 24 h for a period duration of three months starting from January 2020 until March 2020. 

For the development of accurate error sensor compensation of the WSN system, sufficient data for both temperature and irradiance solar data are required for pre–post compensation analysis and Fuzzy rule design. Unique temperature compensation post-processing solutions are achievable when the relation power output of the solar module is constant with the output of the decompensates temperature and irradiance. The oscillating measured temperatures and irradiance are neutralized with a reasonable amount of difference between the minimum and maximum measurement in the Fuzzy compensation rule ranges. In order to achieve these conditions, one month’s data for temperature and irradiance sensor from the real application in the solar module were used for preprocessing and testing the Fuzzy rule compensation of the WSN system. Table 4 shows the measurement of real-time data for the solar PV panel monitoring system. 

Table 4 demonstrates that the real-time result of temperature measurement on the surface PV panel is drifted from the temperature measurement at the back of the PV panel. This error was caused by several factors such as the direct exposure to the sensor on the surface panel and electrical installation condition. The result shows the error between temperature measurements of both sides of the PV panel achieved up to 20%. The range of this error is not acceptable. The high percentage error temperature measurement is on temperature measurement readings above 40 °C. This is caused by the solar PV panel, which absorbed heat that increases the high temperature compared to the surrounding environment temperature, which recorded a maximum daily temperature of 34 °C. The percentage errors in the temperature measurement on the surface PV panel base on the back PV panel reading in real time are 23.64 °C, 36.8 °C, 47.63 °C, and 52.2 °C: 4.01%, 21.19%, 17.4%, and 20.10%, respectively.

Figure 20 shows the display parameters measurement of the PV module on the IoT Thinger platform. Figure 21, Figure 22, Figure 23 and Figure 24 show the graph measurement of the temperature, power PV output, wind speed, and solar irradiance.

### 8.1. Compensation of Temperature Error Measurement Using Fuzzy Logic Scheme

Long-term data were captured where acquisition occurs over 10 days are required to provide adequate temperature data where the data are recorded every minute and saved in the Thinger IO cloud. Analysis of long-period data acquisitions is necessary to gather the required data for accurate temperature compensation post-processing. Applying temperature compensation to the measured temperature sensor that is located at the back of the PV panel will reduce the error of the captured temperature data and allow for higher statistical confidence in the measurement, allow prediction of the PV surface temperature, and improvise the accuracy of power PV output forecasting or monitoring of the PV system. The temperature sensor located at the surface of the PV panel is used as a reference sensor and as an input for the Fuzzy rule scheme. Data capture duration of 24 h for 10 days has been shown to give the required amount and type of data and to allow for a reasonable pace of statistical analysis. Figure 25a–e shows the graph of the surface PV panel temperature and the back panel PV temperature before compensating using the Fuzzy Logic scheme error.

Figure 26a–e displayed the temperature measurement of the PV panel before and after temperature compensation post-processing results for the measurement of the PV panel based on the data collected on 2–7 March 2020. A duration of 120 h temperature data was used where the data were sorted before compensation using the sensor error compensated proposed method. Careful examination of the compensated and uncompensated temperature measurement observed the maximum compensate temperature error of the temperature measurement achieved up to 17, 15, 10, 8, and 10 °C for 3 March, 4 March, 5 March, 6 March, and 7 March, respectively. This compensated value temperature reflects the range of 14% to 27% of temperature that was compensated based on the maximum temperature of 55 °C. The highest percentage of the compensated temperature proved the proposed Fuzzy Logic compensation scheme effectively reduced the error temperature sensor measurement of the PV panel system.

Figure 27 combines the simulation result using the Fuzzy Logic compensation scheme on a Matlab platform. The signal displays the temperature of the surface, back panel PV system, and the output Fuzzy Logic compensation algorithm that represents the surface temperature. The Fuzzy temperature signal output is in the range of the Fuzzy rule that was implemented on the compensation scheme. The compensated temperature showed that the signal is highly accurate to compensate the PV back panel temperature above 40 °C. A large amount of data captured within the before and after datasets results are needed to improve the estimates of the surface PV panel based on the back PV panel temperature, even in low temperatures and other disturbances such as rain. Additionally, evidence of the quality of the performance of the transducer performance for a particular measurement can be extracted visually from these plots, and this information is valuable when deciding on the trustworthiness of the results.

#### Temperature Error Modeling

ANOVA analysis using the response surface method was conducted to establish the thermal sensor error of the PV panel measurement behavior model. The effect of the PV back temperature and irradiance density to the measurement error and the prediction of the surface temperature PV panel was investigated.

ANOVA analysis was performed to check the accuracy of the mathematical models in predicting the PV panel device temperature error. A F ratio for models below 0.05 means that the empirical model represents the system and is ideal for predicting the response of the system, while the *F* value lacking fit for this model was found to be higher than 0.1 to ensure an accurate model design.

Table 5 shows the ANOVA for the Response Surface Quadratic Model of PV Temperature Error measurement and prediction.

The Model F-value of 38.73 implies the model significance. There is only a 0.01% chance that this large “Model F-Value” could happen because of noise. Values less than 0.0500 of “Prob > F” mean that the terms of the model are important. B^2^ are important terms of a model and in this case B. The mathematical modeling of the sensor error measurement for the PV surface panel in terms of coded factors is given by the following equation:(12)$$T_{error}=14.88+8.94B-0.7A^{2}-B^{2}$$
where *A* is the irradiance and *B* is the PV back panel temperature. The effect of the PV error measurement in relation to the PV back temperature and solar irradiance density is clearly shown in the 3-D plot and contour in Figure 28. The graph statistically proposes that the increased temperature of the PV panel is the most significant factor that influences the increase in temperature error measurement, while the solar irradiance density is less significant in the increase percentage error. For the collected data in which the highest PV panel temperature is 52.2 °C, the recorded percentage error on the PV surface temperature measurement is 20.74%. This highest error will affect the prediction power output solar panel system, since the temperature and solar irradiance are the most important factors related to the number of power outputs of the solar farm system.

The interaction relationship between the temperature of the PV back panel and the temperature error for the surface PV panel measurement is shown in Figure 29. Based on the graph, the increase of back PV panel temperature increases the error of measurement of the surface PV panel without the implementation of a sensor error compensation algorithm to the measurement system or to the statistical analysis. The increase in temperature error measurement is significant; it is almost linear for temperatures between 21 and 46.5 °C. The error of PV surface measurement starts to reduce when the PV back panel measurement is higher than 46.5 °C.

It is concluded that the temperature sensor thermal behaviours on the PV solar module are caused by factors such as the solid structure, installation, material properties, heat generation rate, sensor position, working conditions, and ambient temperature, which influence thermal error. The thermal error compensation with strong robustness is necessary to eliminate the complex errors that involve many parameters. Moreover, the empirical finding in laboratory calibration correlation between the thermal error and the value to compensate cannot be simplified, since the laboratory calibration is done under a controlled environment compared with the real implementation on the solar module site influenced by several factors. In rigorous modelling and compensation, the previous method of sensor error compensation that implements an empirical correlation between the thermal error and critical temperature fails, because the thermal error model does not identify the difference zone that is the Fuzzy system rule in this study to be differentiated to three different laws.

The temperature error of the PV surface measurement is directly related to the location of the temperature sensor, drift voltage effect by the length of the sensor cable, installation wiring, etc. An innovative and robust error compensation is needed, and the correction method should reflect the error mechanism and the installation factor. The temperature compensation process should be investigated to ensure a highly accurate implementation of real-time error correction.

### 8.2. Compensation of Irradiance Error Measurement Using Fuzzy Logic Scheme

Figure 30 shows the raw data irradiance measurement obtained from the Si-mV-85 irradiance sensor and SM206 Solar Power Meter, which were collected from the developed WSN based on the IoT platform before being compensated using the Fuzzy Logic compensation scheme. The average irradiance measurement between 3 March and 5 March 2020 using an mV-85 irradiance sensor was 261.69 W/m^2^ and the average solar irradiance during the day-time was 645 W/m^2^. The average maximum solar irradiance measurement over three days was 1235.5 W/m^2^. Careful examinations of the solar irradiance raw data measured by the sensor Si-mV-85 irradiance demonstrated that the percentage error of measurement was between 20% and 31% compared to the SM206 Solar Power Meter. The high percentage error is contributed by several factors such as insufficient experimental data during the calibration process, the effect environmental factors that were not considered during the calibration, and the voltage drift effect that occurred due to the use of a long wired sensor during the installation of the solar PV location as compared to the short cable used during the calibrations process at the laboratory. This factor affects the accuracy of the solar irradiance measurement. Solar irradiance is the most influential factor in the performance of the solar PV system power output, as previous research has proven that the increment of irradiance density is linear to the increase of the PV panel power output. The power PV generation forecasting solar irradiance density and PV surface temperature were used by the researcher to predict the power output. Figure 30 shows the raw data solar irradiance measurement of the Si-mV-85 irradiance sensor and SM206 Solar Power Meter.

Solar irradiance measurements provide valuable information to the power management team in energy utility companies to calculate and manage the power distribution of the solar power plant. This information can also be used to determine the possibilities for retrofitting the power plant. Several stages of a solar power plant require accurate data of solar irradiation. It begins in the early life cycle of a solar power plant, where accurate measurements of solar irradiance allow the project developer to select the most suitable location. For a bigger power plant with a 50 MW peak, each percent in measurement error can mean more than a million US dollars in misjudgment of the project’s total revenue. 

A demonstration of the value of solar irradiance data before and after the compensation using Fuzzy Logic compensation scheme can be observed in Figure 30. The same data are displayed in Figure 31, where the data measured using a SM206 solar meter were removed, and the solar irradiance compensated profile was added to show the benefit and improvement of solar irradiance Fuzzy Logic scheme (SIFLS) compensation. The calculation of the percentage error correction using SIFLS compensated data provides a reduction error between 8.5% and 18.5%. The measurement of solar irradiance density was 350 W/m^2^, and when not using SIFLS, the compensation was 1100 W/m^2^, which can be identified on the 3 March 2020 solar irradiance data profile. Analyses of solar irradiance data on 4 March and 5 March revealed that the maximum percentage compensated by SIFLS was 18.8% and 20.6%, respectively.

There are several commercial solar irradiance sensors available to measure irradiance density. These instruments inherently have different sources and magnitudes of errors due to different designs and operating principles. A previous study by [15] showed that many sensors and transducers were affected by the drift voltage because of the length wiring that caused voltage loss due to the increase in wire resistance. Other factors include loose wire connection and environmental factors such as the wire’s direct exposure to the sun and radioactivity. Possible explanations for this typical behavior of this type of sensor are the existence of dirt and particles settlement on a solar panel sensor that affects its measurement capability. This explanation for these solar irradiance error measurements is substantiated at the time of this study.

A combination of the compensated and uncompensated solar irradiance and the SM206 solar meter measured is shown in Figure 32. The Si-mV-85 irradiance sensor measured uncompensated 310 W/m^2^ on 3 March 2020 at 7:30 a.m. was compared to the 100 W/m^2^ compensated measurement that is the output of SIFLS, which demonstrated an absolute difference of 210 W/m^2^. This deviation represents a 19.4% percentage error of solar irradiance sensor compared to the reference of the SM206 power meter. Table 6 and Table 7 show the percentage of compensation of the sensor Si-mV-85 irradiance measurement on selected acquisition hours for 4 March and 5 March 2020.

The data in Table 6 and Table 7 show the correction percentage of solar irradiance measurement using SIFLS, which tabulates the measurement of the Si-mV-85 sensor, SM206 power meter, and compensate sensor measurement. The data demonstrated a significantly improved confidence measurement for solar irradiance density when SIFLS compensation has been applied to the Si-mV-85 sensor solar irradiance measurement data. For daytime hours between 10:00 a.m. and 15:00 p.m., the SIFLS can reduce the percentage error of solar irradiance measurement between 4% and 23%. This demonstrated the benefit of applying SIFLS compensation to the solar irradiance Si-mV-85 sensor measurement datasets as it has a significant impact in the reduction of the percentage error sensor measurement.

Another explanation of the effect of an error in the measurement of solar irradiance sensors is the inappropriate technical procedure during sensor installation. A review by Abdul Rahim et al. [7] shows that the sensor irradiance error measurement was caused by a tilted and oriented sensor configuration. A study by Vignola [29] also indicated that the calibration process method of a solar irradiance sensor affects the measurement precision. They proposed to conduct the calibration under three cloud conditions to determine the validity of the calibration coefficient for different weather types.

#### Irradiance Error Modeling

Table 8 shows the ANOVA response analysis of irradiance density measurement error. The error of irradiance measurement was identified to be less influenced by temperature. The model was designed for a confidence level of 96.73% with an acceptable reduction in the model. The *F*-value model for the irradiance measurement error was 14.90. The *p*-value of 0.0001 implied that the model is significant with a negligible influence of noise. There is only a 0.01% chance that the model would be affected by noise.

The final equation in terms of coded factors for the percentage error of irradiance measurement is given by:(13)$$I_{error}=28.94+12.22A-7.35A^{2}-2.06B^{2}$$
where *A* is irradiance and *B* is the PV back panel temperature. Figure 33c illustrates interactions between the solar back PV panel temperature and the density irradiance measurement to the percentage error irradiance measurement. The increase in the irradiance value significantly increased the percentage error irradiance measurement. Factors that contributed to the increase in error percentage are inaccurate calibration, which is the calibration process that needs more testing with different irradiance values in different weather conditions. For the reading in between zero and 400 W/m^2^, the increase in percentage error on irradiance measurement increased almost linearly. However, the increase in the percentage error measurement started to reduce for irradiance measurement readings above 500 W/m^2^ until 1400 W/m^2^, as shown in Figure 33. The result indicated that the measurement of solar irradiance density is influenced by many factors that contribute to the error of measurement.

## 9. Conclusions

The main objective of this study is to compensate for the temperature and solar irradiance sensor error measurement of the PV solar system through WSN with the fixed installation. The most important contribution of this research is the realization of sensor error compensation using the intelligent Fuzzy Logic algorithm. The most notable advantage of the proposed method is that it improves the measurement of sensor accuracy and reduces the error of sensor affected by several factors at different working conditions. The temperature and solar irradiance error modelling method was proposed with the support of an analysis of the error mechanism, and the thermal behaviour modelling method was proposed to determine the surface PV system temperature. The WSN installation through the IoT platform was established, and the principles of measurement were introduced. The temperature and irradiance error Fuzzy compensation scheme were also established with the present model in Section 4. Testing of thermal error modeling and compensation methods were performed to analyze the efficiency of the device.

The simulation for the solar irradiance Fuzzy Logic compensation (SIFLC) scheme and Temperature Fuzzy Logic compensation (TFLC) scheme were performed using Matlab/Simulink to validate and compare the proposed scheme effectiveness. The SIFLC and TFLC were firstly simulated in Simulink for testing. The Fuzzy Logic toolbox that was available was used. The Sugeno-Type Fuzzy Inference system was used as the Fuzzy inference engine. For the Fuzzification, the trapezoidal MF was used for real-time implementation. In the Fuzzy logic toolbox, the developed rules can be changed, and the effect was observed easily. The defuzzification of the Sugeno output MF is either linear or constant singleton spikes. Based on the simulation result, the scheme was used to compensate the error temperature and solar irradiance sensor measurements over variation temperature and solar irradiance range from 20 to 60 °C and from 0 up to 2000 W/m^2^. It is shown that the excellent compensation percentage error reduction was achieved by the proposed compensation scheme. From the experimental results, several significant features may be noted:Proposed solar irradiance Fuzzy Logic compensation (SIFLC) and Temperature Fuzzy Logic compensation (TFLC) error scheme to compensate the error of temperature and solar irradiance measurement on PV panel monitoring.Established wireless network sensor through IoT platform for PV solar system.Reduced the solar irradiance error percentage and temperature measurement when the proposed compensation techniques were applied to the sensor measurement data.Initial experimental results suggest that systematic error will reduce significantly since the sensor measurement percentage error reflects the systematic error in the overall statistical data analysis.Improve the precision of sensor measurement when SIFLC and TFLC implemented the sensor measurement system. It is expected that the precision of power forecasting that utilizes the solar irradiance and temperature as the input could increase.It is possible to improve the proposed compensation scheme by combining other intelligent algorithms. This can be achieved through initial data training before simulation with SIFLC and TFLC.The performance achieved in this paper is only for one site PV solar system. Thus, the calibration process should be conducted in different areas and different weather conditions to improve the sensor measurement accuracy. The compensation decisions that are made on the basis of a Fuzzy rule may depart from the correct requirements in practice if the temperature and solar irradiance are not sufficiently significant. It is recommended to develop a system rule based on a full year of data.

## Figures and Tables

**Figure 1 sensors-20-06744-f001:**
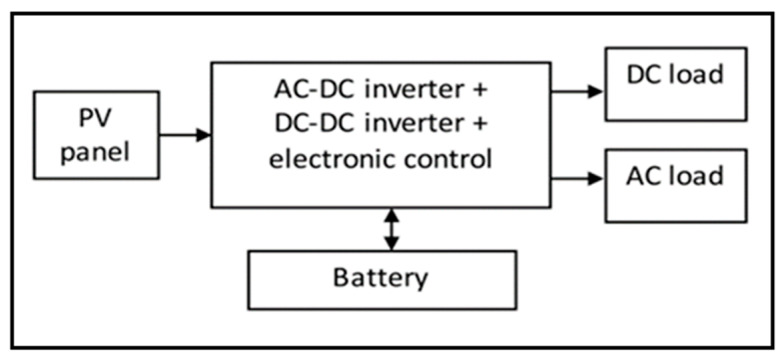
Stand-alone PV solar system block diagram.

**Figure 2 sensors-20-06744-f002:**
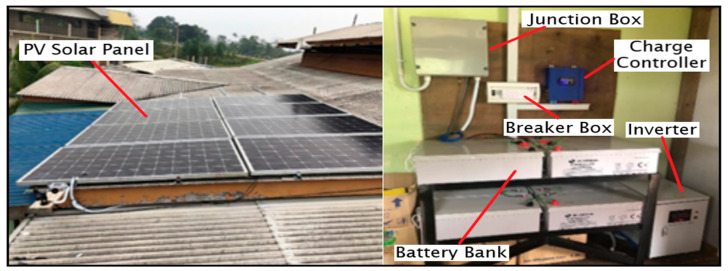
Installation of stand-alone solar system.

**Figure 3 sensors-20-06744-f003:**
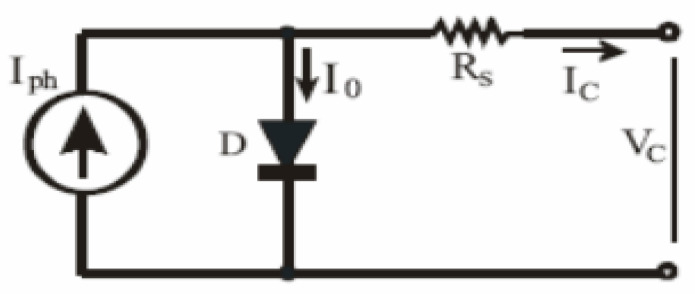
Equivalent circuit of a solar module.

**Figure 4 sensors-20-06744-f004:**
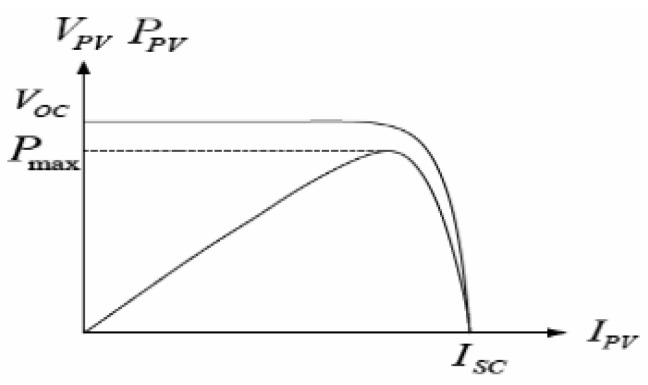
Photovoltaic Current to Voltage (I-V) Characteristic [21].

**Figure 5 sensors-20-06744-f005:**
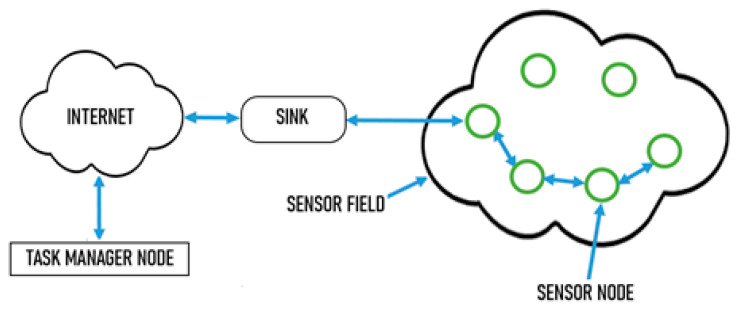
The architecture structure of wireless sensor network based on IoT.

**Figure 6 sensors-20-06744-f006:**
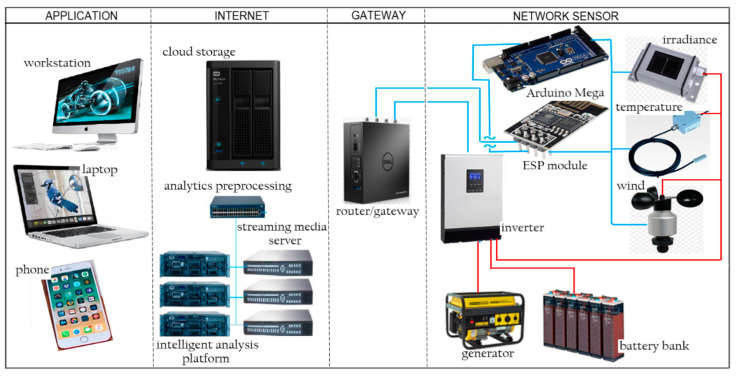
Structure of IoT base wireless sensor network.

**Figure 7 sensors-20-06744-f007:**
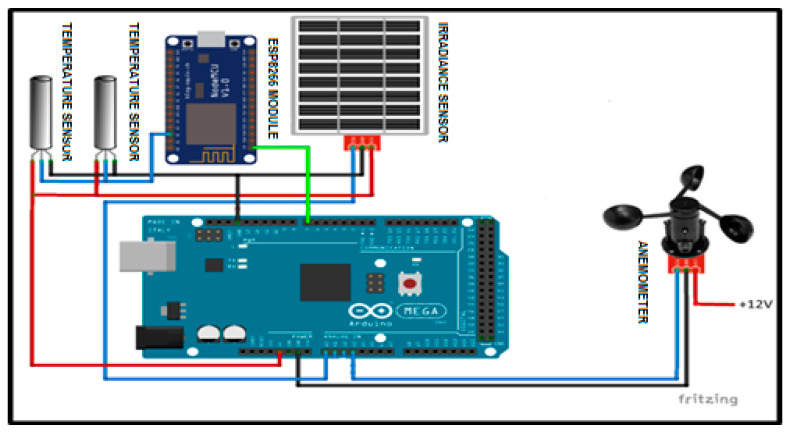
Schematic diagram of the wireless sensor network (WSN) node sensor.

**Figure 8 sensors-20-06744-f008:**
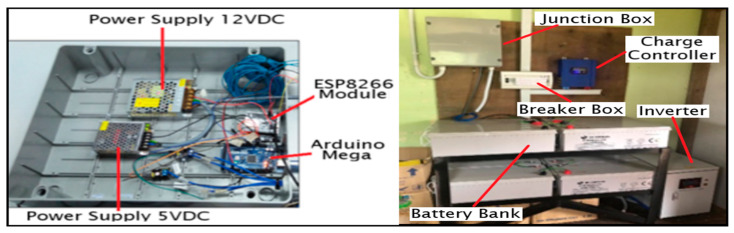
WSN node sensor for the photovoltaic (PV) panel monitoring system.

**Figure 9 sensors-20-06744-f009:**
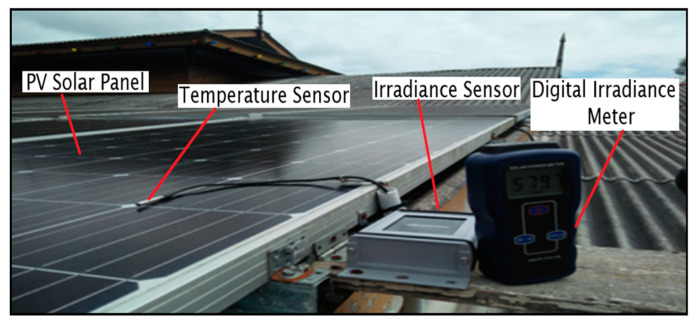
Installation of temperature and solar irradiance sensor for PV module WSN.

**Figure 10 sensors-20-06744-f010:**
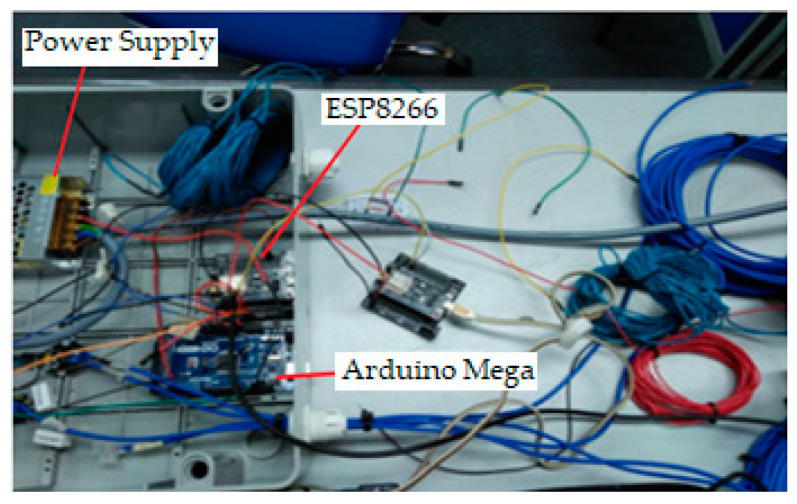
Installation of WSN node sensor based on ESP8266 and Arduino Mega.

**Figure 11 sensors-20-06744-f011:**
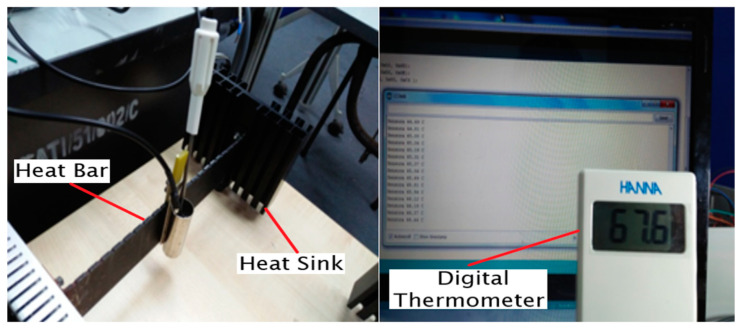
DS18B20 temperature sensor calibration setup.

**Figure 12 sensors-20-06744-f012:**
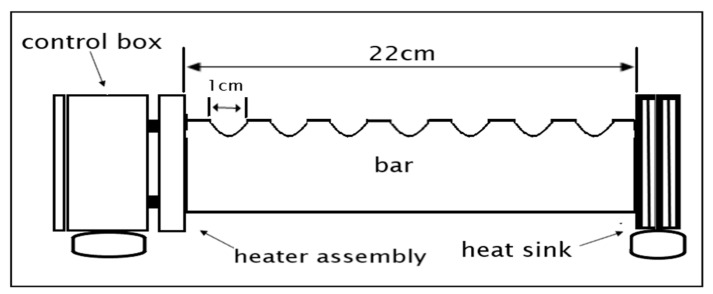
Heat bar schematic diagram.

**Figure 13 sensors-20-06744-f013:**
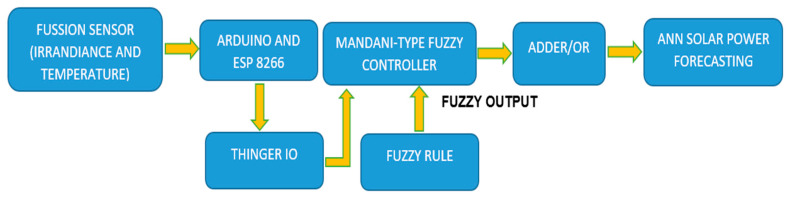
The proposed block diagram for a Fuzzy Irradiance and Temperature Compensation Scheme (FITCS).

**Figure 14 sensors-20-06744-f014:**
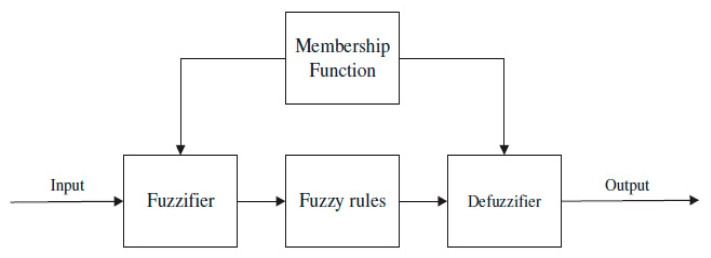
Basic block for Fuzzy Logic.

**Figure 15 sensors-20-06744-f015:**
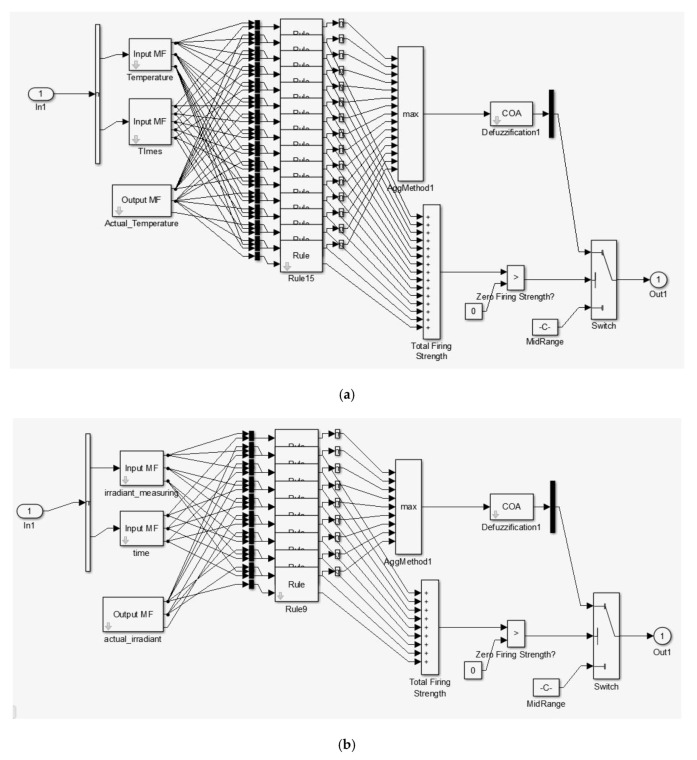
Fuzzy Inference System (FIS) Internal Block Function: (**a**) FIS for Temperature, (**b**) FIS for Irradiance.

**Figure 16 sensors-20-06744-f016:**
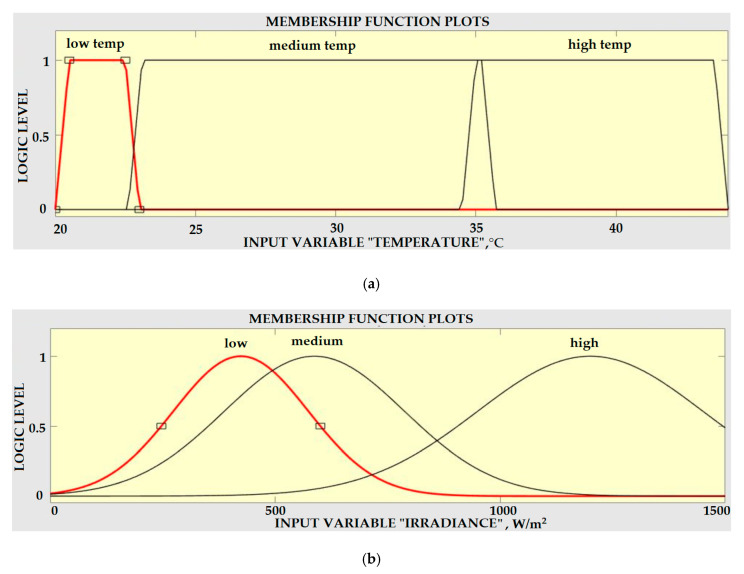
Input Fuzzy sets (**a**) Membership functions for temperature (**b**) Membership functions for irradiance.

**Figure 17 sensors-20-06744-f017:**
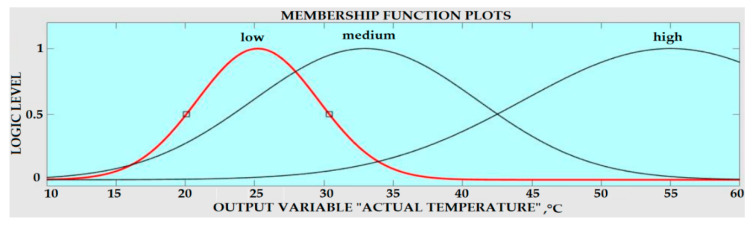
Membership functions for the output Fuzzy.

**Figure 18 sensors-20-06744-f018:**
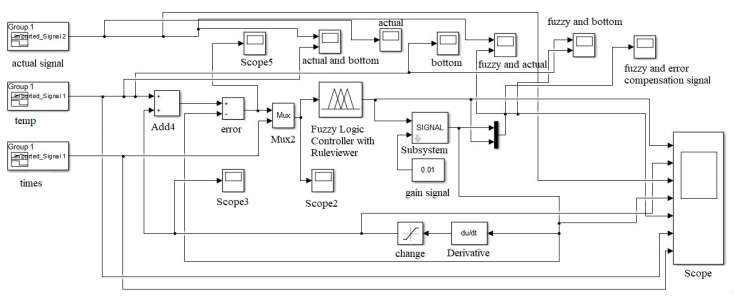
Temperature Fuzzy Simulink error compensation system.

**Figure 19 sensors-20-06744-f019:**
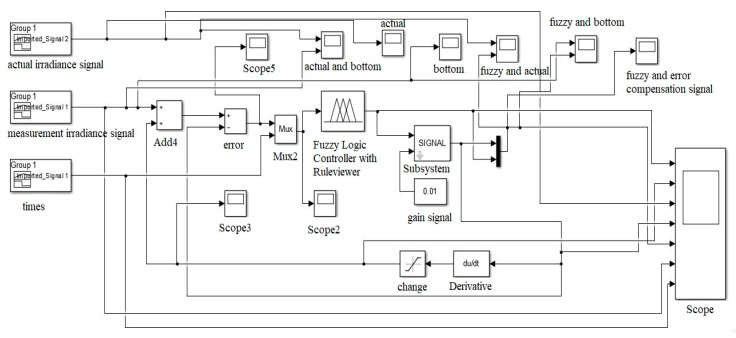
Irradiance Fuzzy Simulink error compensation system.

**Figure 20 sensors-20-06744-f020:**
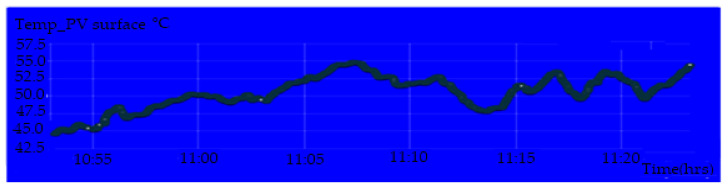
Temperature measurement on the surface of the PV module.

**Figure 21 sensors-20-06744-f021:**
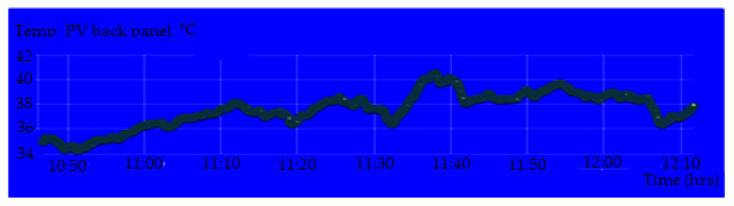
Temperature measurement at the back of the PV module.

**Figure 22 sensors-20-06744-f022:**
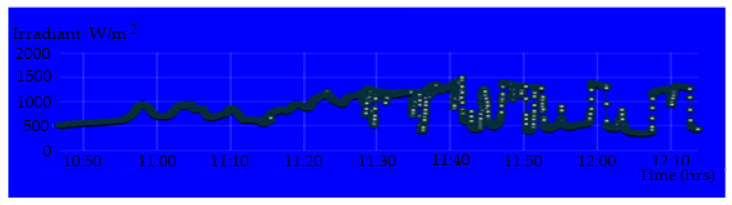
Irradiance measurement at the location of the PV module system.

**Figure 23 sensors-20-06744-f023:**
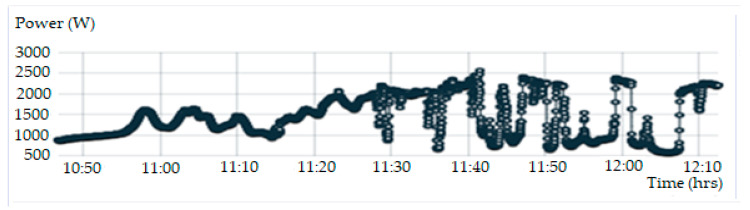
Measurement of the PV power output.

**Figure 24 sensors-20-06744-f024:**
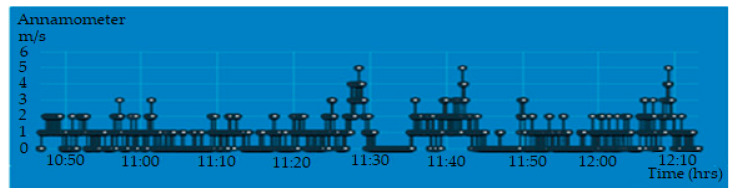
Wind speed measurement at the location of the PV module system.

**Figure 25 sensors-20-06744-f025:**
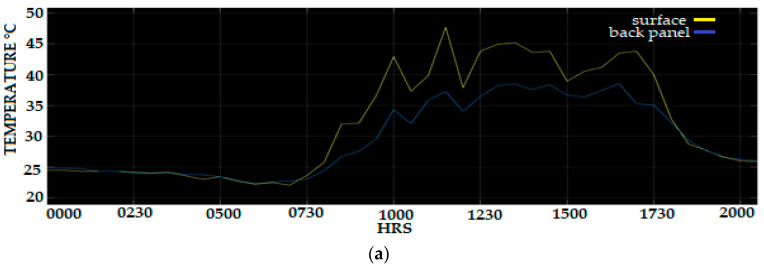
PV surface and back panel temperature (**a**) 3 March (**b**) 4 March (**c**) 5 March (**d**) 6 March (**e**) 7 March.

**Figure 26 sensors-20-06744-f026:**
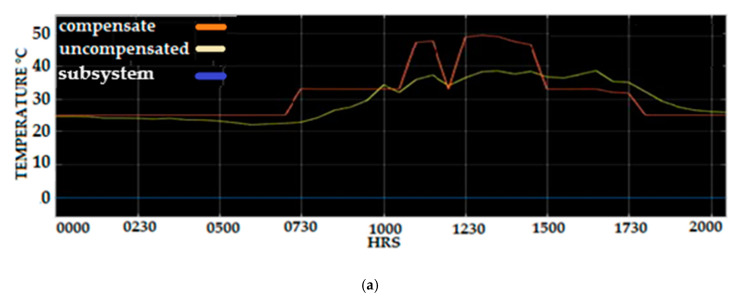
Uncompensated and compensated back PV panel temperature and (**a**) 3 March (**b**) 4 March (**c**) 5 March (**d**) 6 March (**e**) 7 March.

**Figure 27 sensors-20-06744-f027:**
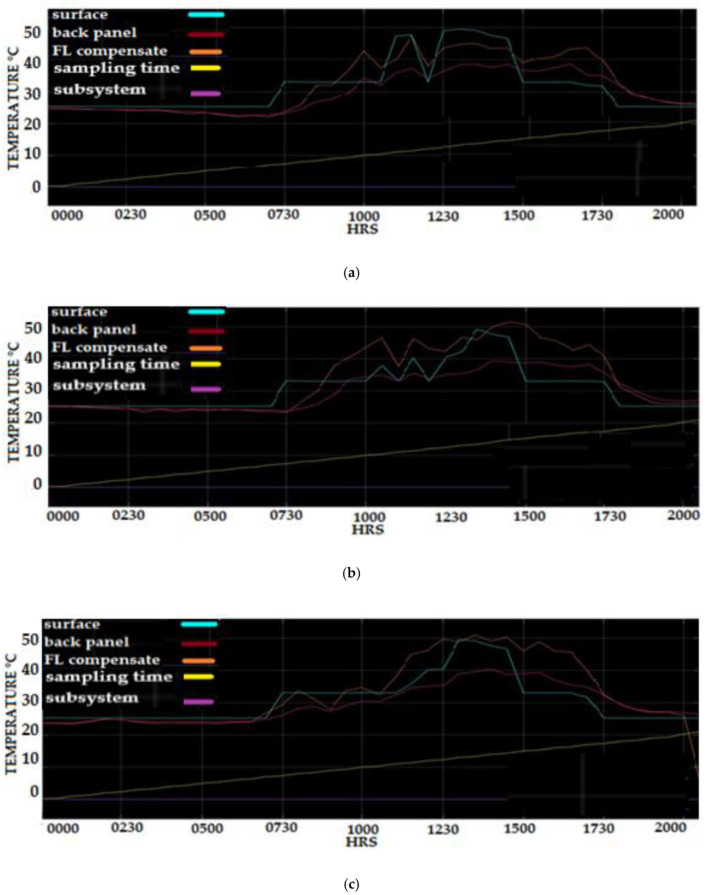
PV panel surface temperature, back PV panel temperature, and the Fuzzy Logic compensate temperature output (**a**) 3 March (**b**) 4 March (**c**) 5 March (**d**) 6 March (**e**) 7 March

**Figure 28 sensors-20-06744-f028:**
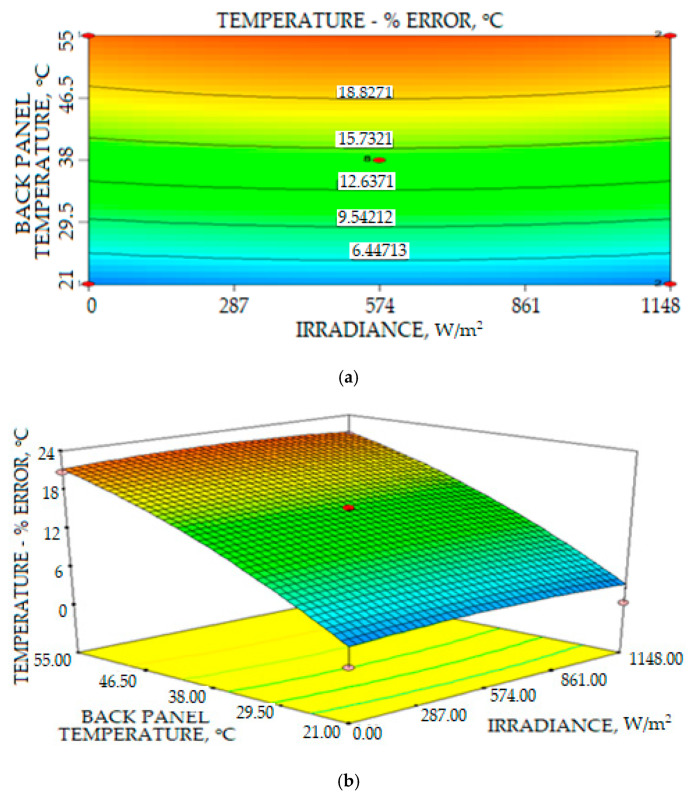
Relations of the PV back panel temperature and irradiance intensity to the error of the PV surface temperature (**a**) Contour plot (**b**) 3D plot.

**Figure 29 sensors-20-06744-f029:**
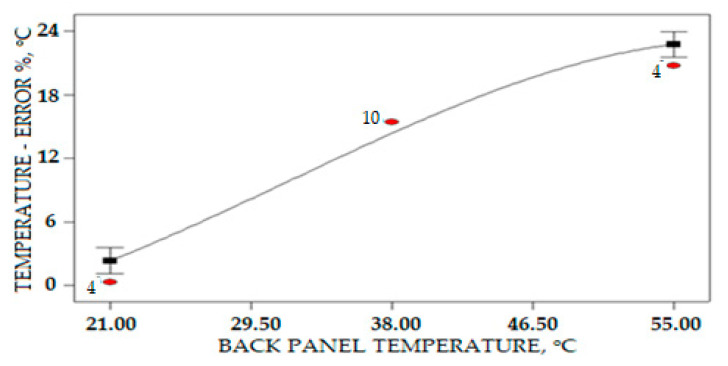
Interaction of the PV surface measurement error with the increase of PV back panel temperature.

**Figure 30 sensors-20-06744-f030:**
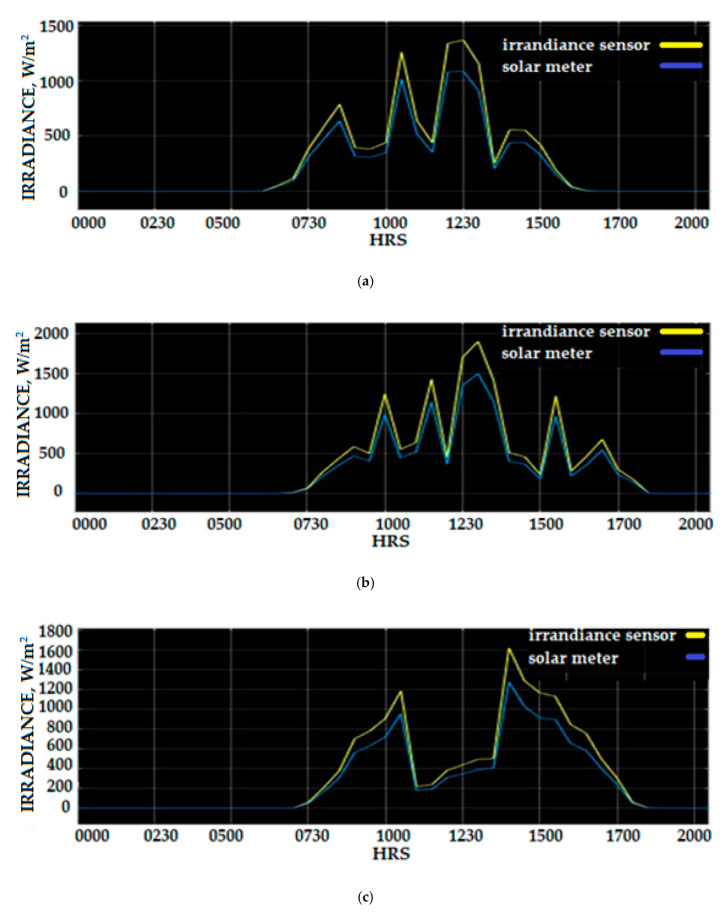
Irradiance measurement of Si-mV-85 irradiance sensor and SM206 Solar Power Meter: (**a**) 3 March 2020 (**b**) 4 March 2020 (**c**) 5 March 2020.

**Figure 31 sensors-20-06744-f031:**
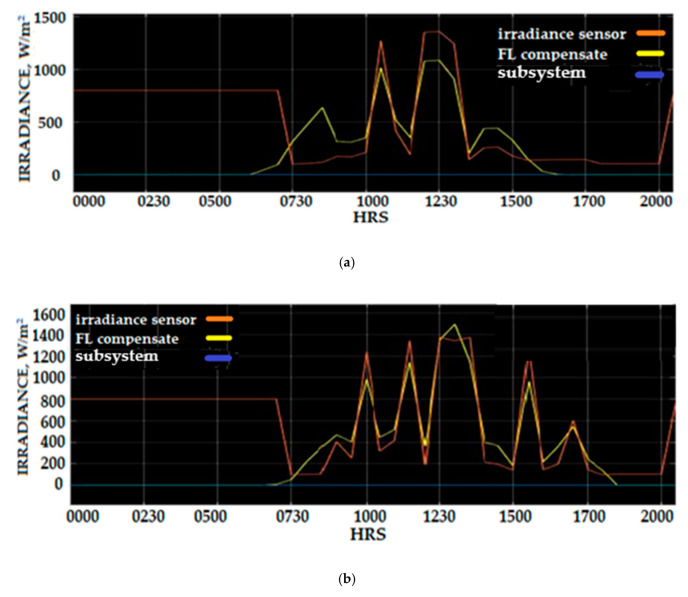
Irradiance measurement of the Si-mV-85 irradiance sensor and compensated irradiance signal using the Fuzzy Logic scheme (**a**) 3 March (**b**) 4 March (**c**) 5 March.

**Figure 32 sensors-20-06744-f032:**
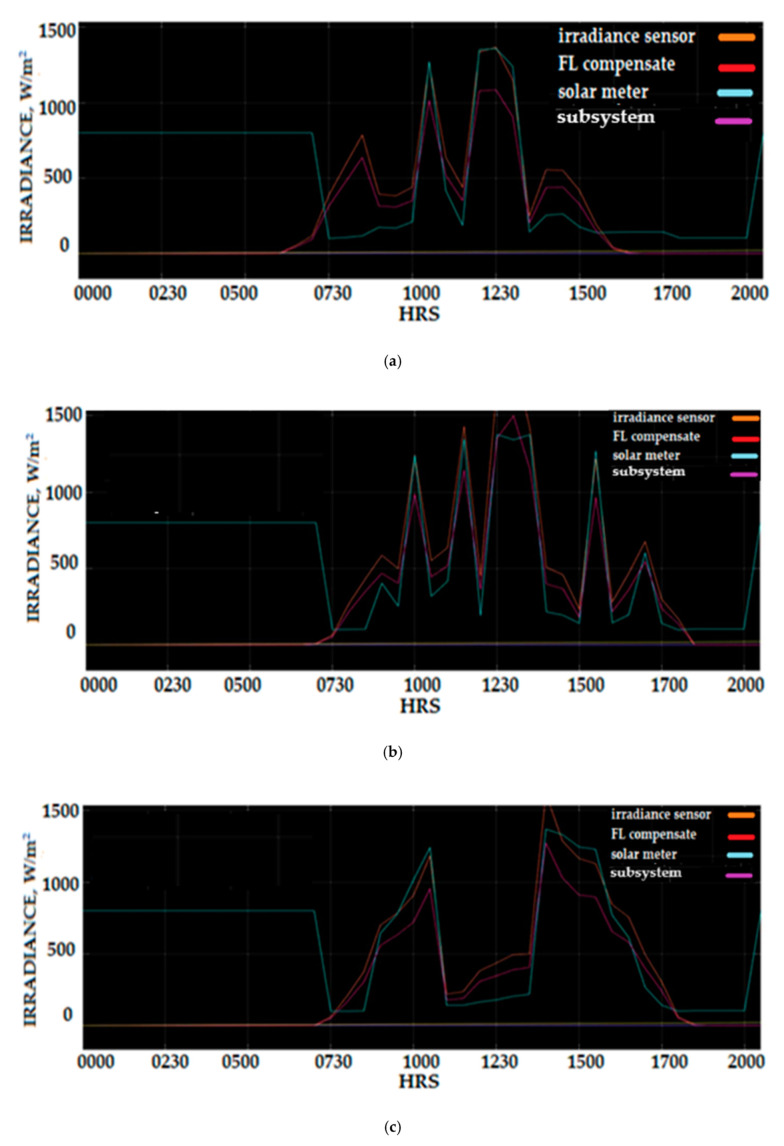
Combination of Fuzzy signal output, irradiance measurement of Si-mV-85, SM206 Solar Power Meter, and compensate irradiance signal using the Fuzzy Logic scheme (**a**) 3 March (**b**) 4 March (**c**) 5 March.

**Figure 33 sensors-20-06744-f033:**
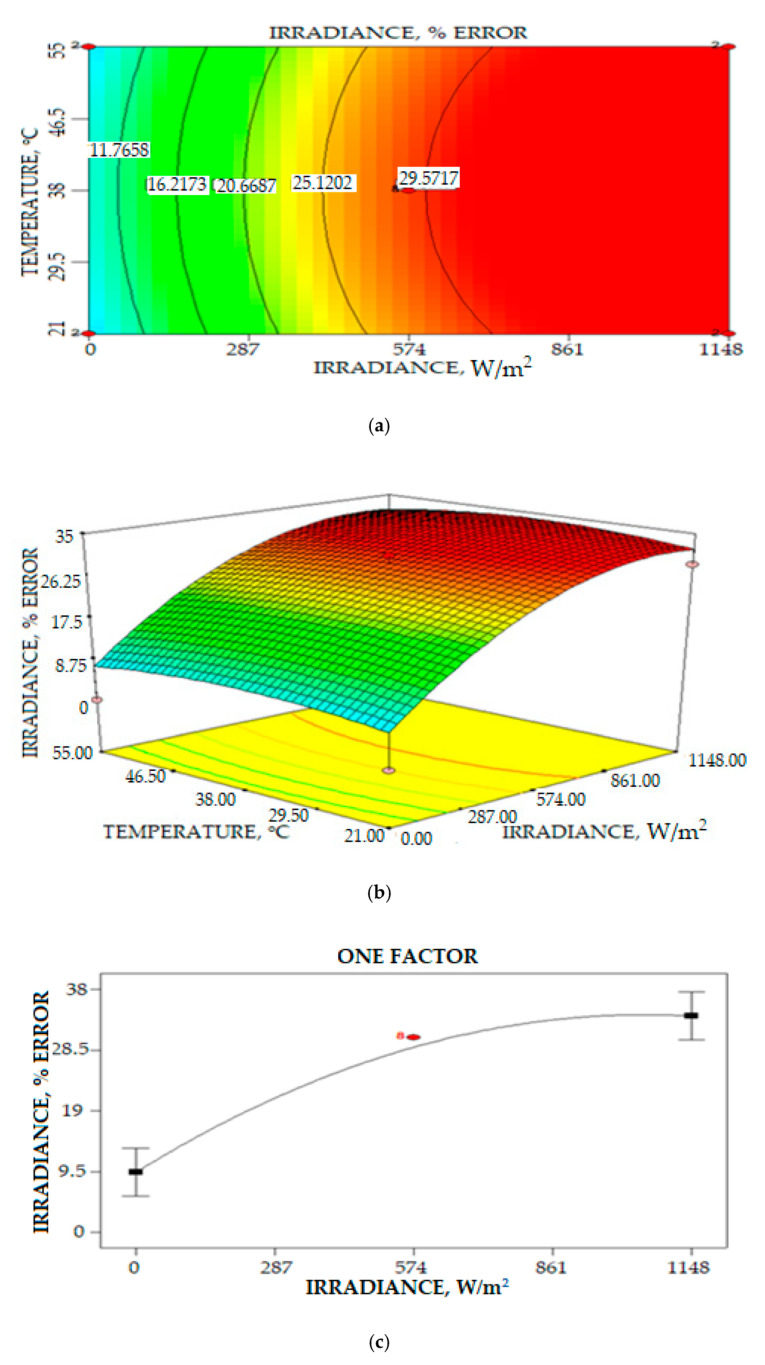
Relations of solar irradiance intensity and temperature to the percentage error of solar irradiance measurement: (**a**) Contour plot (**b**) 3D plot (**c**) One Factor relationship plot.

**Table 1 sensors-20-06744-t001:** BSM340M monocrystalline 330 W electrical parameters.

Parameters	Value
Rated Maximum Power at STC (W)	330
Open Circuit Voltage (Voc/V)	46.34
Maximum Power Voltage (Vmp/V)	37.57
Short Circuit Current (Isc/A)	9.29
Maximum Power Current (I_mp_/A)	8.78
Moduel Effieciency (%)	17.02
Power Tolerance (W)	0–5
Temperature Coefficient of I_sc_	+0.058%/°C
Temperature Coefficient of V_oc_	−0.330%/°C
Temperature Coefficient of P_max_	−0.410%/°C
STC	Irradiance 1000 W/m^2^, Cell Temperature 25 °C, Air Mass 1.5

**Table 2 sensors-20-06744-t002:** Absolute error and percentage error of irradiance calibration measurement.

Reference Value (W/m^2^)		Variation	
Commercial Meter	Measure Sensor	Absolute Error	Percentage Error, %
885	631	254	28.7
768	590	178	23.18
834	580	254	30.46
796	565	231	29.02
660	480	180	27.27
598	420	178	29.77
528	372	156	29.55
495	365	130	26.26
474	325	149	31.43
462	319	143	30.95
453	310	143	31.57
441	306	135	30.61
447	315	132	29.53
416	298	118	28.37
410	294	116	28.29
406	289	117	28.82
412	300	112	27.18
421	312	109	25.89
434	319	115	26.50
446	325	121	27.13
		Average error	28.52%

**Table 3 sensors-20-06744-t003:** Absolute error and percentage error of temperature calibration measurement.

Reference Value (°C)		Variation	
$T_{b}$	$T_{u}$	Absolute Error	Percentage Error, %
36.00	33.31	2.69	7.47
35.69	32.56	3.13	8.76
33.50	32.00	1.50	4.48
34.75	33.00	1.75	5.04
31.06	29.40	1.66	5.35
30.06	28.00	2.06	6.86
44.50	35.63	8.88	19.94
38.19	34.19	4.00	10.47
42.00	35.25	6.75	16.07
40.13	34.94	5.19	12.93
37.81	33.81	4.00	10.58
39.63	34.38	5.25	13.25
39.19	34.06	5.13	13.08
38.69	34.00	4.69	12.12
37.13	32.94	4.19	11.28
36.88	33.25	3.63	9.83
35.13	33.19	2.25	6.35
40.81	37.00	3.81	9.34
43.88	37.06	6.81	15.53
		Average error	10.32%

**Table 4 sensors-20-06744-t004:** Recorded data for PV panel monitoring system on 1 March 2020.

Date and Time	Wind (m/s)	Irradiant (W/m^2^)	Power (W)	T_PVsurface_ (°C)	T_PVback_ (°C)
1 March 2020 T00:30	0	0	0	22.1	22.75
1 March 2020 T01:00	0	0	0	22.3	22.63
1 March 2020 T01:30	1	0	0	22	22.19
1 March 2020 T02:00	1	0	0	21.2	21.44
1 March 2020 T02:30	1	0	0	21	21.25
1 March 2020 T03:00	1	0	0	21	20.88
1 March 2020 T03:30	1	0	0	21.2	20.94
1 March 2020 T04:00	0	0	0	21.2	20.88
1 March 2020 T04:30	0	0	0	21.31	21.13
1 March 2020 T05:00	0	0	0	23	22.19
1 March 2020 T05:30	0	0	0	21.88	21.88
1 March 2020 T06:00	0	0	0	21.31	21.19
1 March 2020 T07:00	0	3.5	6	21.88	21.81
1 March 2020 T07:30	0	157.5	270	23.64	22.69
1 March 2020 T08:00	1	287	492	27.06	24.38
1 March 2020 T08:30	0	427	732	32.5	26.43
1 March 2020 T09:00	2	574	984	36.38	29
1 March 2020 T09:30	1	311.5	534	41.13	31.88
1 March 2020 T10:00	1	259	444	39.75	33.63
1 March 2020 T10:30	1	329	564	44.69	34.56
1 March 2020 T11:00	3	350	600	43.44	35.94
1 March 2020 T11:30	3	1148	1968	43.94	36.13
1 March 2020 T12:00	3	1123.5	1926	48.82	39.13
1 March 2020 T12:30	3	1144	1962	49.06	39.31
1 March 2020 T13:00	2	1057	1812	50.13	39.5
1 March 2020 T13:30	1	1078	1848	52.12	41.31
1 March 2020 T14:00	1	966	1656	49.38	39.31
1 March 2020 T14:30	1	906.6	1554	45.25	38.31
1 March 2020 T15:00	2	815.5	1398	47.63	39.32
1 March 2020 T15:30	1	682	1170	43.06	38.31
1 March 2020 T16:00	1	640	1098	43.94	39
1 March 2020 T16:30	1	479.5	822	44.5	38.38
1 March 2020 T17:00	2	360.5	618	43.69	37.81
1 March 2020 T17:30	1	213	366	28.94	35.56
1 March 2020 T18:00	1	42	72	31.31	32.19
1 March 2020 T18:30	0	0	0	27.25	28.88
1 March 2020 T19:00	2	0	0	26.63	26.88
1 March 2020 T19:30	1	0	0	25.94	26.06
1 March 2020 T20:00	0	0	0	25	25.12
1 March 2020 T20:30	0	0	0	24.4	24.81
1 March 2020 T21:00	0	0	0	24.1	24.44
1 March 2020 T21.30	0	0	0	23.8	24
1 March 2020 T22.00	0	0	0	23.5	23.88
1 March 2020 T22.30	0	0	0	23.2	23.5
1 March 2020 T23.00	0	0	0	23.1	23
1 March 2020 T23.30	0	0	0	23.1	22.8
1 March 2020 T24.00	0	0	0	23.0	22.8

**Table 5 sensors-20-06744-t005:** ANOVA for the Response Surface Quadratic Model of Temperature Error.

	Sum of		Mean		*p*-Value	
Source	Squares	df	Square	*F*-Value	Prob > F	
Model	1147.18	5	229.44	38.73	<0.0001	significant
A-Irradiance	0.000	1	0.000	0.000	1.0000	
B-Temp	1090.62	1	1090.62	184.12	<0.0001	significant
AB	0.000	1	0.000	0.000	0.0400	significant
A^2^	7.07	1	7.07	1.19	0.0292	significant
B^2^	52.50	1	52.50	8.86	0.0100	significant
Residual	82.93	14	5.92			
Lack of Fit	82.93	3	27.64	2.45		not significant
Pure Error	0.000	11	0.000			
Cor Total	1230.11	19				

**Table 6 sensors-20-06744-t006:** Solar irradiance Fuzzy Logic scheme (SIFLS) percentage compensation error for 4 March solar irradiance.

Time	Si-mV-85	SM206	Fuzzy Scheme_compensated_	% Error	% Compensate Error
10:00	1000	1240	1250	20.00	0.81
12:30	1305	1555	1375	16.07	11.57
15:00	185	245	195	24.48	20.40

**Table 7 sensors-20-06744-t007:** SIFLS percentage compensation error for 5 March solar irradiance.

Time	Si-mV-85	SM206	Fuzzy Scheme_compensated_	% Error	% Compensate Error
10:00	690	885	1000	22.03	12.99
12:30	325	475	435	31.5	8.52
15:00	880	1165	1235	24.46	6.08

**Table 8 sensors-20-06744-t008:** ANOVA for response surface quadratic model of irradiance error.

	Sum of		Mean		*p*-Value	
Source	Squares	df	Square	*F* Value	Prob > F	
Model	2854.44	5	570.89	14.90	<0.0001	significant
A-IRRADIANCE	2040.55	1	2040.55	53.26	<0.0001	
B-TEMP	0.000	1	0.000	0.000	1.0000	
AB	0.000	1	0.000	0.000	1.0000	
A^2^	785.41	1	785.41	20.50	0.0005	
B^2^	61.57	1	61.57	1.61	0.2256	
Residual	536.43	14	38.32			
Lack of Fit	536.43	3	178.81			
Pure Error	0.000	11	0.000			
Cor Total	3390.87	19

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
