# Peer review of "Design and Implementation of Fuzzy Compensation Scheme for Temperature and Solar Irradiance Wireless Sensor Network (WSN) on Solar Photovoltaic (PV) System"

_sensors, 2020, doi:10.3390/s20236744_

Round 1

Reviewer 1 Report

This article has everything that need to be a good article, but it need to be shorter and highlighting the strength points. I consider that can be cut a lot from all sections.

I consider that the article is very long and the reader at one moment it is loosing the interest.

For an example in Section 2, it is explained even how it is composed a solar panel. This things are well-known, and I consider that it is not necessary to insist, because this article suppose to be address to people who are knowing this thing.

Please use instead of "20oC to 60oC", 20ºC to 60ºC.

Please use instead of "2000Wm2", 2000 Wm2.

Please use instead of Amps just A. Example: row 154, "8.78 Amps "

Row 287, I believe that it is a typo mistakes.."222VAC"

Figure 8, 19-the writing is not legible(readable). Please use a greater font.

Rows 261 and 312, I believe that it is a typo mistakes at the name of the temperature sensor

The authors are pleased to pay attention at the text, because are typos mistakes.

To be comparable the" Temperature measurement on surface PV module. ", from Figure 20, it supposed to have the same time interval like Figure 21.

Author Response

No

Comments

Remarks

1

This article has everything that need to be a good article, but it need to be shorter and highlighting the strength points. I consider that can be cut a lot from all sections.

I consider that the article is very long and the reader at one moment it is loosing the interest.

Noted. We try to reduce and make it more compact article.

2

For an example in Section 2, it is explained even how it is composed a solar panel. This things are well-known, and I consider that it is not necessary to insist, because this article suppose to be address to people who are knowing this thing.

Yes Prof. We have delete some paragraph in section 2 which explain general knowledge/information.

3

Please use instead of "20oC to 60oC", 20ºC to 60ºC.

Yes Prof. We have make correction on the format.

4

Please use instead of "2000Wm2", 2000 Wm2

Noted and revised as per commented.

5

Please use instead of Amps just A. Example: row 154, "8.78 Amps "

Noted and revised as per commented.

6

Row 287, I believe that it is a typo mistakes.."222VAC"

Noted and revised as per commented.

7

Figure 8, 19-the writing is not legible(readable). Please use a greater font.

Noted and revised as per commented.

8

Rows 261 and 312, I believe that it is a typo mistakes at the name of the temperature sensor

Yes Prof. We have make correction for the typo mistake

9

The authors are pleased to pay attention at the text, because are typos mistakes

Yes Prof. We have check it and revised the typo error.

10

To be comparable the" Temperature measurement on surface PV module. ", from Figure 20, it supposed to have the same time interval like Figure 21.

Figure 20 and 21 shows the display data in our system. The comparable data between surface and back panel PV system shown in results section 8.

Reviewer 2 Report

- The article deals with the measurement of solar radiation intensity and temperature of PV panels on photovoltaic power plants. These parameters are important for the management of photovoltaic power plants and therefore I think that the article is suitable for Sensors journal.

- The introduction talks about the possibilities of replacing diesel electric generators with photovoltaic power plants. I think that the need for electricity accumulation should also be mentioned, because the PV power plant gives no power during the night. Sometimes during the day, it can be cloudy and the power of the PV power plant can be low. For example, the article (Poulek, V., Dang, M.Q., Libra, M., Beránek, V., Šafránková, J., PV Panel with Integrated Lithium Accumulators for BAPV Applications - One Year Thermal Evaluation. IEEE Journal of Photovoltaics, 2020, 10(1), 150-152, ISSN 2156-3403, DOI: 10.1109/JPHOTOV.2019.2953391) deals with the energy storage from the photovoltaic power plant and it could be mentioned.

- The article could be reduced, for example Fig. 1 shows a well-known fact and is therefore unnecessary.

- Throughout the text, symbols of physical quantities should be written in italica font. Physical unit symbols are correctly written in the standard font.

- Temperature units should be written „°C“, not „oC“ (line 22, 124, 164, tab.1 and so on).

- Line 23, 916 – The unit of irradiance should be W.m-2, not Wm2.

- There should be spaces before the reference number (for example line 36, 42, 53, 67, 69, 96, 200, 236 and so on). If more references in one place, they should be in one bracket (for example line 81 [9,10], line 200 [12,13], line 448 [6, 16] and so on.

- Line 242 - The unit of irradiance should be kW.m-2 or kW/m2 not kW/m2. (Line 222, 223, 242, 329, 330 and so on).

- There should be spaces between value and unit (for example line 301, 369)

- Standard symbols of units should be used – A, not Amps (line 151, 154), W not watt (line 102)

- Figs.17, 18 - The horizontal axis should be signed by the quantity and the relevant unit.

- Figs. 20, 21, 22, 23, 24 - The horizontal axis should be signed „Time (h)“.

- Figs. 25, 26, 27, 30, 31, 32– The axes are not signed, the numbers on scales are not readable.

- Figs. 28, 29, 33 - The axes should be signed by the quantity and the relevant unit.

My recommendation - accept after major revisions.

Author Response

No

Comments

Remarks

1

The article deals with the measurement of solar radiation intensity and temperature of PV panels on photovoltaic power plants. These parameters are important for the management of photovoltaic power plants and therefore I think that the article is suitable for Sensors journal.

Noted with many thanks Prof.

2

The introduction talks about the possibilities of replacing diesel electric generators with photovoltaic power plants. I think that the need for electricity accumulation should also be mentioned, because the PV power plant gives no power during the night. Sometimes during the day, it can be cloudy and the power of the PV power plant can be low. For example, the article (Poulek, V., Dang, M.Q., Libra, M., Beránek, V., Šafránková, J., PV Panel with Integrated Lithium Accumulators for BAPV Applications - One Year Thermal Evaluation. IEEE Journal of Photovoltaics, 2020, 10(1), 150-152, ISSN 2156-3403, DOI: 10.1109/JPHOTOV.2019.2953391) deals with the energy storage from the photovoltaic power plant and it could be mentioned.

Yes, agreed with your suggestion, Prof. We have added the important of energy storage PV system in the introduction section.

3

The article could be reduced, for example Fig. 1 shows a well-known fact and is therefore unnecessary.

Yes Prof. We have reduced the article especially in section 2.

4

Throughout the text, symbols of physical quantities should be written in italica font. Physical unit symbols are correctly written in the standard font.

Noted and revised the symbols as commented.

5

Temperature units should be written „°C“, not „oC“ (line 22, 124, 164, tab.1 and so on).

Noted and revised the units as commented.

6

Line 23, 916 – The unit of irradiance should be W.m-2, not Wm2.

Noted and revised the units as commented.

7

There should be spaces before the reference number (for example line 36, 42, 53, 67, 69, 96, 200, 236 and so on). If more references in one place, they should be in one bracket (for example line 81 [9,10], line 200 [12,13], line 448 [6, 16] and so on.

Yes Prof. We have make correction and combine the bracket for more references.

8

Line 242 - The unit of irradiance should be kW.m-2 or kW/m2 not kW/m2. (Line 222, 223, 242, 329, 330 and so on).

Noted and revised the units as commented.

9

There should be spaces between value and unit (for example line 301, 369)

Yes Prof. We have make correction for the format

10

Standard symbols of units should be used – A, not Amps (line 151, 154), W not watt (line 102)

Noted and revised the units as commented.

11

Figs.17, 18 - The horizontal axis should be signed by the quantity and the relevant unit.

Noted and revised as per commented.

12

Figs. 20, 21, 22, 23, 24 - The horizontal axis should be signed „Time (h)“.

Noted and revised as per commented.

13

Figs. 25, 26, 27, 30, 31, 32– The axes are not signed, the numbers on scales are not readable.

Noted and revised as per commented.

14

Figs. 28, 29, 33 - The axes should be signed by the quantity and the relevant unit.

Noted and revised as per commented.

Round 2

Reviewer 1 Report

It is much better than before. 

Congratulation on your work!

Reviewer 2 Report

The comments were accepted and errors were corrected.

Still in the figs. 28, 30, 31, 32, 33, the irradiance units are bad, there should be W/m2, not Wm2.

I suggest accepting the article after correcting the above mentioned figures.